# Policy Contrastive Decoding for Robotic Foundation Models

**Shihan Wu**[1*]  **Xu Luo**[1*]   **Ji Zhang**[2†]  **Junlin Xie**[1]
**Jingkuan Song**[3]   **Heng Tao Shen**[3]   **Lianli Gao**[1]
[1]University of Electronic Science and Technology of China
[2]Southwest Jiaotong University   [3]Tongji University
shihan.wu.koorye@outlook.com,jizhang.jim@gmail.com

https://koorye.github.io/PCD

## Abstract

Robotic foundation models, or generalist robot policies, hold immense potential to enable flexible, general-purpose and dexterous robotic systems. Despite their advancements, our empirical experiments reveal that existing robot policies are prone to learning *spurious correlations* from pre-training trajectories, adversely affecting their generalization capabilities beyond the training data. To tackle this, we propose a novel **Policy Contrastive Decoding (PCD)** approach, which redirects the robot policy's focus toward object-relevant visual clues by contrasting action probability distributions derived from original and object-masked visual inputs. As a training-free method, our PCD can be used as a *plugin* to improve different types of robot policies without needing to finetune or access model weights. We conduct extensive experiments on top of three open-source robot policies, including the autoregressive policy `OpenVLA` and the diffusion-based policies `Octo` and $\pi_0$. The obtained results in both simulation and real-world environments prove PCD's flexibility and effectiveness, e.g., PCD enhances the state-of-the-art policy $\pi_0$ by **8.9**% in the simulation environment and by **108**% in the real-world environment.

## 1 Introduction

Recent efforts in developing flexible, general-purpose, and dexterous robotic systems have largely focused on robotic foundation models, or generalist robot policies. The goal of such policies is to empower users to instruct robots to perform arbitrary tasks, enabling autonomous task execution with minimal human supervision. Specifically, the policy takes as input a visual observation of the robot's state combined with a language instruction that defines the task, and outputs a robot action, such as end-effector displacement. Ongoing advancements in scaling real-world robotic data corpora have facilitated the development of numerous robot policies (Brohan et al., 2022; Kim et al., 2024; M. et al., 2024; Pertsch et al., 2025), which have demonstrated exceptional effectiveness in controlling various robots across diverse environments and acquiring a wide range of manipulation skills.

Despite their notable achievements, existing robot policies are prone to learning *spurious correlations* (Ye et al., 2024; Geirhos et al., 2020; Liu et al., 2023a) from pre-training trajectories, adversely affecting their ability to generalize outside their training data. As illustrated in Fig. 1, the robot policy predominantly relies on spurious features (e.g., background or textures) rather than object features within observations to predict actions (see **(a)(d)**). Consequently, slight alterations to the visual observation background, like adjustments to the panning light region (i.e., **(a)→(b)**) and the handle position (i.e., **(a)→(c)**), lead to **36**% and **32**% drops in the policy's action prediction performance, respectively. Prior work in other fields has demonstrated that models relying on spurious correlations often suffer significant performance degradation on test data when a distribution shift occurs between the training and test environments (Lu et al., 2025; Varma et al., 2024; Izmailov et al., 2022). This

---
[*]Equal contribution
[†]Corresponding author

underscores the importance of redirecting robot policies' focus from spurious features to object-relevant ones to enhance their generalization across different scenarios during inference. Therefore, we provide a first study on the following question:

> *Can we propose a training-free, plug-and-play approach to mitigate the adverse effects of spurious correlations, thereby rectifying predicted actions for existing robot policies?*

In this work, we present a Policy Contrastive Decoding (**PCD**) approach to answer the question. The core idea of our PCD approach is to shift the robot policy's attention from spurious features to object-relevant ones by contrasting action probability distributions derived from original and object-masked observation images during inference. To ensure both effectiveness and flexibility of the approach, a *Tracking-to-Mask (Track2Mask)* strategy is introduced, which first annotates target-specific objects in the *initial* visual observation using human expertise (e.g., Point and Box prompts) or off-the-shelf object detection models (e.g., Grounding DINO (Liu et al., 2024b)), then employs the SAM2 (Kirillov et al., 2023) model to track these objects across subsequent observations in the trajectory. This enables precise object masking with minimal or no human intervention. Moreover, unlike autoregressive policies, diffusion-based policies cannot directly generate action probability distributions to feed into our PCD. To cope with this limitation, we introduce a *KDE-based Probabilistic Modeling (KDE-PM)* scheme, which calculates action probability distributions for diffusion-based policies using the kernel density estimation (KDE) (Wkeglarczyk, 2018) technique. This enables our PCD approach to be compatible with both autoregressive and diffusion-based policies.

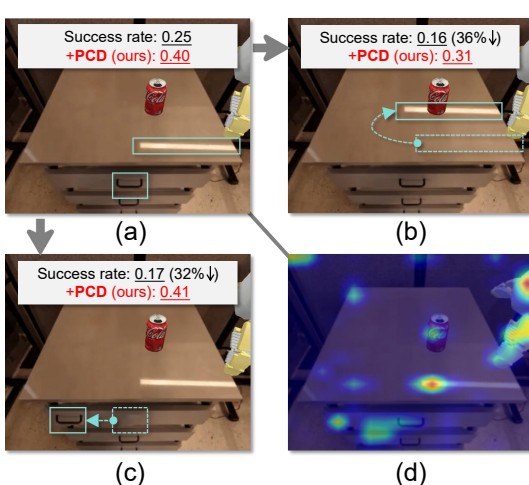

To the best of our knowledge, this is the first work to introduce a training-free approach for resolving the spurious correlation issue in robot policies. We conduct extensive evaluations in both simulation and real-world environments, across a total of 15 diverse tasks, and on top of three open-source robot policies—including the autoregressive policy `OpenVLA` (Kim et al., 2024) and the diffusion-based policies `Octo` (M. et al., 2024) and $\pi_0$ (Black et al., 2024). The achieved results demonstrate the strong flexibility and effectiveness of our PCD approach. On simulation benchmarks, PCD enhances the performance of `Octo`, `OpenVLA` and $\pi_0$ by **29.7**%, **50.6**% and **8.9**%, respectively. On real-world manipulation tasks, PCD outperforms the state-of-the-art robot policy $\pi_0$ by **108**%.

Figure 1: **Robot policies tend to spuriously correlate task-irrelevant features with actions, compromising their ability to generalize to unseen scenarios.** As observed, changing the light position from **(a)** to **(b)** and the drawer handle position from **(a)** to **(c)** results in **36**% and **32**% drops in the performance of the baseline policy `OpenVLA` (Kim et al., 2024), respectively. **(d)** Attention map. More results are in Section 4.4 and **Appendix** A.

**Contributions**. Our contributions in this work are threefold. **1**) We propose PCD, a simple, training-free, and easy-to-implement scheme to tackle the spurious correlation issue in robot policies. **2**) PCD can be used as a plugin to enhance both autoregressive and diffusion-based policies, without requiring fine-tuning or access to pre-trained model weights. **3**) Extensive experiments demonstrate PCD's effectiveness and flexibility, consistently improving three state-of-the-art policies across 15 diverse tasks.

## 2 RELATED WORK

**Robotic Foundation Models**. The rapid development of robotic foundation models or generalist robot policies has been significantly inspired by the success of large visual and language models (Karamcheti et al., 2024a; Zhai et al., 2023a; Touvron et al., 2023a). Robot policies can be categorized based on action prediction strategies into autoregressive models and diffusion-based policies. Autoregressive policies sequentially generate action tokens based on previous outputs (Brohan et al., 2022; 2023; Zawalski et al., 2024; Belkhale et al., 2024; Zhang et al., 2024). `OpenVLA` (Kim et al.,

2024) equips with a Llama 2 (Touvron et al., 2023a) language model alongside a visual encoder that merges pretrained features from SigLIP (Zhai et al., 2023a) and DINO-v2 (Oquab et al., 2024). Diffusion-based policies simultaneously generate entire action dimensions by performing diffusion processes on noise (M. et al., 2024; Liu et al., 2024a). `Octo` (Team et al., 2024) learns a lightweight policy on the OXE dataset (Collaboration et al., 2024), enabling rapid adaptation to unknown tasks and scenarios on standard consumer-grade GPUs. $\pi_0$ (Black et al., 2024) exhibits flexible control over different robot types by leveraging pre-trained general knowledge from vision-language models. Although most robot policies are trained on extensive robotic behavior datasets, recent studies have shown that slight variations in deployment scenarios can significantly affect their generalization performance on downstream tasks in real-world environments (Firoozi et al., 2023; Gao et al., 2024).

**Contrastive Decoding**. Recent studies in large vision-language models (VLMs) have highlighted the challenge of *hallucination* (Gunjal et al., 2024; Rawte et al., 2023; Liu et al., 2023b; Zhang et al., 2025), where models generate inaccurate or misleading outputs not grounded in the input data. Contrastive Decoding (CD) (Favero et al., 2024; Chen et al., 2024; Leng et al., 2024) suppresses hallucinations by comparing output distributions from original and distorted input without altering the model's weights. Instruction Contrastive Decoding (ICD) (Wang et al., 2024) is used to combat hallucinations in multimodal tasks by modulating the confidence in multimodal alignment of the model's visual and textual inputs, enabling it to distinguish between hallucinated and relevant tokens. Visual Contrastive Decoding (VCD) (Park et al., 2025; Favero et al., 2024) aims to reduce object hallucinations by comparing outputs from original and distorted visual inputs. This approach provides a computationally efficient solution as it requires neither additional training nor external pre-trained models. Our proposed approach in this work is mainly inspired by the success of VCD schemes. However, unlike those schemes that contrast output distributions from original and noise-distorted (Park et al., 2025) or image-removed (Favero et al., 2024) inputs to counteract LVLMs' overreliance on language priors, our PCD contrasts action probability distributions derived from original and object-masked inputs, thereby redirecting the robot policy's focus to object-relevant visual features and enhancing their generalization across different scenarios.

## 3 PROPOSED APPROACH

In this section, we present PCD, our training-free, plug-and-play scheme designed to tackle the spurious correlation issue for existing generalist robot policies. Notably, PCD does not require access to the pre-trained parameters of these robot policies; i.e., it treats each robot policy as a *black-box*.

### 3.1 PRELIMINARIES

This work focuses on robot manipulation tasks where the objective is to leverage a learned language-conditioned policy $\pi_\theta(\boldsymbol{a}_i \mid \boldsymbol{o}_i, \ell)$ to predict an $M$-dimensional action $\boldsymbol{a}_i = [a_1, ..., a_M] \in \mathbb{R}^M$ conditioned on the current visual observation $\boldsymbol{o}_i$ and the language instruction $\ell$ of the task. In this part, we formalize two mainstream paradigms for action prediction: autoregressive action prediction (e.g., `OpenVLA` (Kim et al., 2024)) and diffusion-based action prediction (e.g., `Octo` (M. et al., 2024) and $\pi_0$ (Black et al., 2024)).

**Autoregressive Robot Policies**. The policy decomposes the action prediction task into sequential token generation. Each action dimension $a_t$ of $\boldsymbol{a}_i$ is sampled autoregressively from the probability distribution, conditioned on the visual observation $\boldsymbol{o}_i$, the language instruction $\ell$, as well as the previously predicted dimensions $a_{<t} = \{a_t\}_{t=1}^{t-1}$:

$$a_t \sim \pi_\theta\left(a_t \mid \ell, \boldsymbol{o}_i, a_{<t}\right).$$

For example, `OpenVLA` first encodes $\boldsymbol{o}_i$ through a hybrid vision backbone (DINOv2 (Caron et al., 2023) w/ SigLIP (Zhai et al., 2023b) ) and tokenizes $\ell$ using a pre-trained language model (Llama 2 (Touvron et al., 2023b)). These multimodal features are fused in the LLM's embedding space, which then autoregressively emits $M$ discrete tokens. Each token is finally mapped to an action dimension $a_t$ via a de-tokenizer (Kim et al., 2024).

**Diffusion-based Robot Policies**. The policy frames action generation as an iterative denoising process (Ho et al., 2020). Denote $\boldsymbol{e}_i$ the multimodal embeddings fusing $\boldsymbol{o}_i$ and $\ell$. Starting from Gaussian noise $\boldsymbol{a}_i^K \sim \mathcal{N}(\boldsymbol{0}, \mathbf{I})$, the reverse diffusion process refines all the $M$ dimensions of the

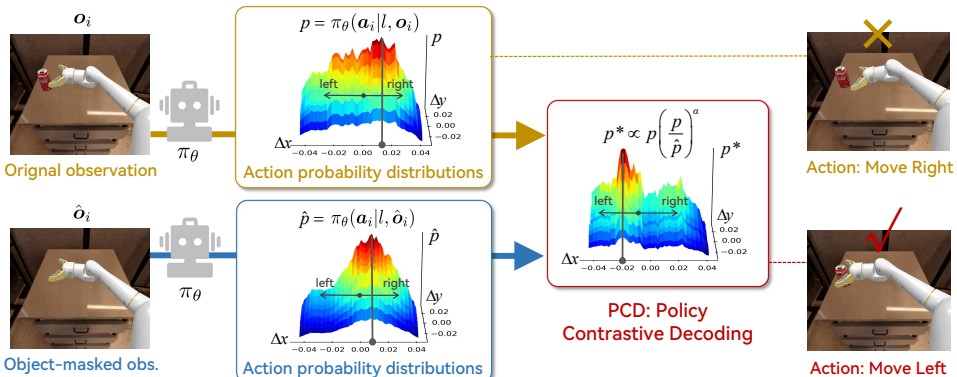

Figure 2: **Overview of our proposed Policy Contrastive Decoding (PCD) approach**. PCD serves as a *plugin* to redirect the robot policy's focus toward object-relevant visual cues by contrasting action probability distributions derived from original observations $p$ and object-masked observations $\hat{p}$. For illustrative purposes, we visualize the predictions only in the $\Delta x$ and $\Delta y$ dimensions of the robot action space $[\Delta x, \Delta y, \Delta z, \text{rot}_x, \text{rot}_y, \text{rot}_z, \text{gripper}]$.

action $\boldsymbol{a}_i$ in parallel over $K$ steps through:

$$\boldsymbol{a}_i^{k-1} = \alpha \left( \boldsymbol{a}_i^k - \gamma \epsilon_\theta \left( \boldsymbol{a}_i^k, \boldsymbol{e}_i, k \right) \right) + \mathcal{N}(\boldsymbol{0}, \sigma^2 \mathbf{I}) \tag{1}$$

where $\boldsymbol{a}_i^k$ is the noisy action at diffusion step $k$, $\epsilon_\theta \left( \boldsymbol{a}_i^k, \boldsymbol{e}_i, k \right)$ is the denoising network, and $\alpha, \gamma, \sigma$ are parameters of a cosine noise schedule (Nichol and Dhariwal, 2021). By jointly optimizing the entire $M$ action dimensions of $\boldsymbol{a}_i$, diffusion-based robot policies enable fine-grained sampling from the joint probability $\pi_\theta(\boldsymbol{a_i}|l, \boldsymbol{o}_i)$ of all action dimensions.

## 3.2 POLICY CONTRASTIVE DECODING (PCD)

The core idea of our proposed PCD method is to eliminate the adverse effect of spurious correlations by redirecting the robot policy's focus toward object-relevant features during inference. To facilitate the introduction of our PCD, we define spurious correlations as follows.

> **Definition 3.1 (Spurious Correlations)**: *Let observations $\boldsymbol{o}$ and language instructions $l$ be generated by a set of $L$ underlying latent factors $\{c_j\}_{j=1}^{L}$. These factors are decomposed into task-relevant factors, denoted by the random variable $\boldsymbol{u}$, which exclusively determine the ground-truth expert policy such that $p(\boldsymbol{a}|\boldsymbol{o}, l) = p(\boldsymbol{a}|\boldsymbol{u})$, and task-irrelevant factors, denoted by the random variable $\boldsymbol{v}$, which have no causal effect on the expert's actions $\boldsymbol{a}$. Spurious correlation is then defined as the statistical dependence between the action $\boldsymbol{a}$ and these task-irrelevant factors $\boldsymbol{v}$ observed under the training distribution $p_{\text{train}}(\boldsymbol{a}, \boldsymbol{v})$, quantified by the mutual information $I_{\text{train}}(\boldsymbol{a}, \boldsymbol{v})$.*

Spurious correlations, indicated by mutual information $I_{\text{train}}(\boldsymbol{a}, \boldsymbol{v}) > 0$ in the training set, allow a robot policy $\pi_\theta$ to infer action $\boldsymbol{a}$ from task-irrelevant factors $\boldsymbol{v}$. These factors act as shortcuts. However, such correlations are unreliable. Under *out-of-distribution* (OOD) conditions—stemming from environmental shifts or compounding robotic errors—$\boldsymbol{v}$ may change independently of task-relevant factors $\boldsymbol{u}$ and, consequently, $\boldsymbol{a}$, leading to inaccurate action predictions.

A direct approach to mitigate learning from spurious correlations involves filtering out $\boldsymbol{v}$ and utilizing only $\boldsymbol{u}$. Nevertheless, identifying and neutralizing spurious features is challenging due to their significant variability across tasks and scenarios, coupled with interdependencies within the feature space. Drawing inspiration from Vision Contrastive Decoding (VCD) (Park et al., 2025; Favero et al., 2024), which effectively reduces hallucinations in vision-language models, we introduce Policy Contrastive Decoding (PCD). PCD aims to shift the robot policy's focus from spurious features $\boldsymbol{v}$ towards task-relevant features $\boldsymbol{u}$ associated with the target object. This is achieved by contrasting model outputs derived from original visual inputs with those from object-masked visual inputs.

**Method Overview**. Fig. 2 presents an overview of our PCD approach. Suppose we have a pretrained policy $\pi_\theta$. Given the current visual observation $\boldsymbol{o}_i$ and the language command $\ell$, we first generate two

distinct action probability distributions of $\boldsymbol{a}_i$: one is $\pi_\theta(\boldsymbol{a}_i|l, \boldsymbol{o}_i)$ conditioned on the *original* visual input $\boldsymbol{o}_i$, and the other is $\pi_\theta(\boldsymbol{a}_i|l, \hat{\boldsymbol{o}}_i)$ conditioned on *object-masked* counterpart $\hat{\boldsymbol{o}}_i$ that removes all task-relevant factors related to target objects. Then, a new contrastive action probability distribution $\pi_\theta^*(\boldsymbol{a}_i|l, \boldsymbol{o}_i)$ is obtained by contrasting $\pi_\theta(\boldsymbol{a}_i|l, \boldsymbol{o}_i)$ and $\pi_\theta(\boldsymbol{a}_i|l, \hat{\boldsymbol{o}}_i)$:

$$\pi_\theta^*(\boldsymbol{a}_i|l, \boldsymbol{o}_i) = \frac{1}{C} \cdot \pi_\theta(\boldsymbol{a}_i|l, \boldsymbol{o}_i) \left( \frac{\pi_\theta(\boldsymbol{a}_i|l, \boldsymbol{o}_i)}{\pi_\theta(\boldsymbol{a}_i|l, \hat{\boldsymbol{o}}_i)} \right)^\alpha, \tag{2}$$

where $C$ is a normalization constant and a larger value of $\alpha \geq 0$ indicates a stronger amplification of differences between the two distributions. Particularly, $\alpha = 0$ recovers the baseline policy. In such a manner, the obtained $\pi_\theta^*(\boldsymbol{a}_i|l, \boldsymbol{o}_i)$ amplifies model predictions on object-relevant features in $\boldsymbol{u}$, thus exhibiting insensitivity to spurious features in $\boldsymbol{v}$, as illustrated in Fig. 2.

To guarantee the flexibility and practical applicability of our PCD method, we need to address the following two questions:

> *1) How can object-masked observations $\hat{\boldsymbol{o}}_i$ be automatically generated for each trajectory?*
> *2) How can we approximate the policy distribution $\pi_\theta$ for diffusion-based policies?*

**Tracking-to-Mask**. We overcome the first question by devising a *Tracking-to-Mask (Track2Mask)* strategy, which enables precise object masking for sequential visual observations along each trajectory, requiring minimal or no human intervention. Concretely, the first step of Track2Mask is to annotate the task-specified objects in the *initial* visual observation. Inspired by visual prompting techniques (Wan et al., 2024), one can annotate the target objects in the initial observation using Point or Box prompts. Besides, we provide the option of leveraging off-the-shelf (open vocabulary) object detection models, such as Grounding DINO (Liu et al., 2024b) for automatic object annotation. Next, the SAM2 (Kirillov et al., 2023) model is used to track and segment the target object across subsequent observations in the trajectory, and the segmented objects are then inpainted to obtain object-masked observations. For more details of Track2Mask, please refer to **Appendix** B.

**KDE-based Probabilistic Modeling**. We address the second question using *KDE-based Probabilistic Modeling (KDE-PM)*, which approximates action probability distributions for diffusion-based policies through kernel density estimation (KDE) (Wkeglarczyk, 2018). Specifically, we first use the pretrained diffusion-based policy to sample $N$ candidate action predictions: $\{\boldsymbol{a}_i(j)\}_{j=1}^N$, where $\boldsymbol{a}_i(j) = [a_1(j), ..., a_M(j)] \sim \pi_\theta(\boldsymbol{a}_i|l, \boldsymbol{o}_i)$. With the assumption that all action dimensions are independent, i.e., $\pi_\theta(\boldsymbol{a}_i|l, \boldsymbol{o}_i) = \prod_{t=1}^M \pi_\theta(a_t|l, \boldsymbol{o}_i)$, we calculate an approximated probability distribution for each action dimension of the action $\boldsymbol{a}_i$ via KDE:

$$\pi_\theta(a_t|l, \boldsymbol{o}_i) \approx \frac{1}{C'} \sum_{j=1}^N \mathcal{K} \left( \frac{a_t - a_t(j)}{b} \right), \; t = 1, ..., M, \tag{3}$$

where $C'$ is a normalization constant and $\mathcal{K}(\cdot)$ indicates a Gaussian kernel $\mathcal{K}(u) = \frac{1}{\sqrt{2\pi}} e^{-\frac{u^2}{2}}$, $b$ is the bandwidth parameter controlling the smoothness of the distribution, and $\{a_t(j)\}_{j=1}^N$ are the $N$ candidate actions for the $t$-th dimension of the action $\boldsymbol{a}_i$, generated from the diffusion process. Similarly, we can derive an action probability distribution for $a_t$ conditioned on object-masked $\hat{\boldsymbol{o}}_i$, i.e., $\pi_\theta(a_t|l, \hat{\boldsymbol{o}}_i)$. With the independence assumption, from $\pi_\theta(a_t|l, \boldsymbol{o}_i)$ and $\pi_\theta(a_t|l, \hat{\boldsymbol{o}}_i)$ we can compute the joint action distributions, which are then used to compute the contrastive action probability $\pi_\theta^*(\boldsymbol{a}_i|l, \boldsymbol{o}_i)$ using Eq. (2). In this way, our PCD is compatible with diffusion-based robot policies.

**Inference with PCD**. As a *training-free* method, our proposed PCD can be employed as a plugin to improve the inference stage of different types of robot policies (i.e., autoregressive policies and diffusion-based policies). Specifically, for a pre-trained robot policy $\pi_\theta(\boldsymbol{a}_i \mid \boldsymbol{o}_i, \ell)$, given the current observation $\boldsymbol{o}_i$, we can obtain the object-masked $\boldsymbol{o}_i$ (i.e., $\hat{\boldsymbol{o}}_i$) leveraging the devised Track2Mask strategy. Based on $\boldsymbol{o}_i$, $\hat{\boldsymbol{o}}_i$ and a language command $\ell$, the autoregressive policy directly computes contrastive action probability distributions for deriving $\boldsymbol{a}_i$ (w/ Eq. (2)), whereas the diffusion-based policy requires performing the KDE-PM process on its output predictions to achieve the same goal. Pseudocode for our PCD method is provided in **Algorithm 1**.

---

**Algorithm 1** PCD: Policy Contrastive Decoding

---

**Require:** A pre-trained robot policy: $\pi_\theta(\boldsymbol{a}_i \mid \boldsymbol{o}_i, \ell)$; initial observation $\boldsymbol{o}_0$; language command $\ell$; maximum time step $S$.

1: $s \leftarrow 0$
2: **while** $s \leq S$ **do**
3:      Obtain object-masked $\boldsymbol{o}_s$: $\hat{\boldsymbol{o}}_s$                                 ▷ w/ Track2Mask
4:      Obtain $\pi_\theta(\boldsymbol{a}_s|l, \boldsymbol{o}_s)$ and $\pi_\theta(\boldsymbol{a}_s|l, \hat{\boldsymbol{o}}_s)$             ▷ w/ KDE-PM for diffusion-based policies
5:      Compute $\pi_\theta^*(\boldsymbol{a}_s|l, \boldsymbol{o}_s)$ using Eq. (2)
6:      Sample $\boldsymbol{a}_s$ from $\pi_\theta^*(\boldsymbol{a}_s|l, \boldsymbol{o}_s)$ and execute $\boldsymbol{a}_s$
7:      $\boldsymbol{o}_{s+1} \leftarrow$ New observation                           ▷ Obtain a new observation
8:      $s \leftarrow s + 1$
9: **end while**

---

Table 1: **SIMPLER Performance**. Task-specific objects in the initial observation are annotated by artificial Point and Box prompts or the automatic detection results of GDINO (Liu et al., 2024b). The results are the success rate and the 95% confidence interval of 300 trials. **As a plug-and-play approach, PCD consistently enhances the three policies by large margins over the 9 tasks.**

| | Tasks | OpenVLA (Kim et al., 2024) | | | | Octo (M. et al., 2024) | | | | $\pi_0$ (Black et al., 2024) | | | |
| --- | --- | --- | --- | --- | --- | --- | --- | --- | --- | --- | --- | --- | --- |
| | | Base | +PCD | | | Base | +PCD | | | Base | +PCD | | |
| | | | Point | Box | GDINO | | Point | Box | GDINO | | Point | Box | GDINO |
| | **Average** | 16.8±1.4 | 24.4±1.6 | 22.9±1.6 | **25.3**±1.6 | 13.8±1.3 | 17.6±1.4 | 17.4±1.4 | **17.9**±1.4 | 63.9±1.8 | 68.1±1.8 | 68.6±1.8 | **69.6**±1.7 |
| | | | +45.2% | +36.3% | +50.6% | | +27.5% | +26.1% | +29.7% | | +6.6% | +7.4% | +8.9% |
| Google Robot | Close Drawer | 47.3±5.6 | 63.7±5.4 | 63.7±5.4 | **73.3**±5.0 | **31.0**±5.2 | 26.0±5.0 | 27.0±5.0 | 21.0±4.6 | 75.7±4.9 | 74.7±4.9 | **80.3**±4.5 | 75.0±4.9 |
| | Move Near | 58.7±5.6 | **64.0**±5.4 | 62.0±5.5 | 59.0±5.6 | 3.7±2.1 | 6.7±2.8 | 6.0±2.7 | **9.0**±3.2 | 67.3±5.3 | 66.7±5.3 | 68.0±5.3 | **72.3**±5.1 |
| | Open Drawer | 23.3±4.8 | **36.3**±5.4 | 33.0±5.3 | 34.7±5.4 | 0.3±0.7 | 0.3±0.7 | 0.3±0.7 | **1.0**±1.1 | 38.0±5.5 | 47.5±5.7 | 50.3±5.7 | **56.3**±5.6 |
| | Pick Coke Can | 18.0±4.3 | 38.0±5.5 | 43.3±5.6 | **45.3**±5.6 | 29.3±5.2 | **51.3**±5.7 | 50.7±5.7 | 50.7±5.7 | 84.0±4.1 | 84.3±4.1 | 87.3±3.8 | **88.0**±3.7 |
| | Apple Drawer | 0.0±0.0 | 0.3±0.7 | **1.0**±1.1 | 0.7±0.9 | **0.0**±0.0 | **0.0**±0.0 | **0.0**±0.0 | **0.0**±0.0 | 17.0±4.3 | 26.0±5.0 | 26.7±5.0 | **27.3**±5.0 |
| WidX. | Carrot Plate | 0.0±0.0 | **7.3**±2.9 | 0.0±0.0 | 4.7±2.4 | 12.0±3.7 | **22.0**±4.7 | 19.0±4.4 | 20.7±4.6 | 58.0±5.6 | **67.7**±5.3 | 50.0±5.5 | 59.7±5.6 |
| | Eggpl. Basket | 0.3±0.7 | **5.3**±2.5 | 1.7±1.4 | **5.3**±2.5 | 38.3±5.3 | 33.0±5.3 | 31.3±5.2 | **38.7**±5.5 | 86.0±3.9 | 83.7±4.2 | 84.7±4.1 | **87.0**±3.8 |
| | Spoon Towel | 0.0±0.0 | 2.3±1.7 | 1.0±1.1 | **4.3**±2.3 | 9.3±3.3 | 15.0±4.0 | **18.7**±4.4 | 16.0±4.1 | 80.7±4.5 | **86.3**±3.9 | 83.7±4.2 | 84.0±4.1 |
| | Stack Cube | **3.7**±2.1 | 2.7±1.8 | 0.0±0.0 | 0.0±0.0 | 0.0±0.0 | **4.3**±2.3 | 4.0±2.2 | 3.7±2.1 | 68.7±5.2 | 76.3±4.8 | 76.7±4.8 | **77.0**±4.8 |

## 4 EXPERIMENTS

In this section, we seek to answer the following questions:

> 1) Can PCD improve robot policies in both simulation and real-world environments?
> 2) How does the performance vary with different design choices?
> 3) What kinds of spurious correlations can our PCD tackle?

We answer the first question in Sections 4.1 and 4.2, the second question in Section 4.3, and the third question in Section 4.4. Detailed descriptions of the baseline robot policies and the evaluation tasks used in simulation and real-world environments are presented in **Appendix** C.

### 4.1 SIMULATION EXPERIMENTS

We perform simulation experiments in SIMPLER (Li et al., 2024), a real-to-sim evaluation environment designed specifically for real-robot policies. SIMPLER accurately reflects real-world performance, enabling reliable assessment of robotic policies. We evaluate 9 tasks across two robot platforms: 5 tasks using the Google Robot and 4 tasks using the WidowX arm.

**Experimental Setup**. We conduct experiments on top of three diverse robot policies, including the autoregressive policy OpenVLA (Kim et al., 2024), and the diffusion-based policies Octo (base) (M. et al., 2024) and $\pi_0$ (Black et al., 2024). We treat these policies as black-boxes, applying PCD solely based on the output action probability distributions for OpenVLA and the output actions for Octo and $\pi_0$. For our PCD, we annotate the target object in the initial observation using three manners: artificial Point and Box prompts, and automatic detection results from Grounding DINO (Liu et al., 2024b). The number of the sampled noise vectors $N$ in KDE-PM is set to 24 for Octo and $\pi_0$. We perform ablation studies on the hyperparameter $\alpha$ in Eq. (2), as well as on the object detection strategies and object inpainting paradigms of the devised Track2Mask module in Section 4.3.

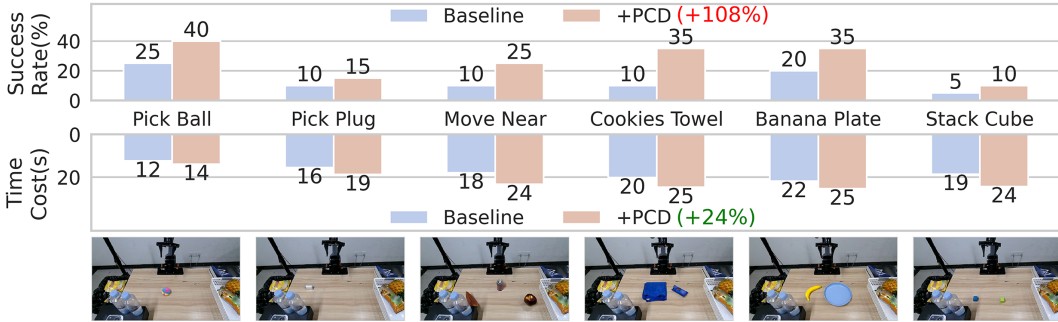

Figure 3: **Real-world Performance**. The target objects in the initial observation are automatically annotated by Grounding DINO (Liu et al., 2024b). **PCD delivers a remarkable 108% performance improvement on the baseline, though it incurs a 24% increase in time cost.**

**Results and Analysis**. Table 1 reports the performance of three robot policies (i.e., `OpenVLA` (Kim et al., 2024), `Octo` (M. et al., 2024) and $\pi_0$ (Black et al., 2024)) integrated w/ or w/o our PCD across a total of 9 tasks from Google Robot and WidowX. From the results in the table, we have the following key observations. **1)** Our PCD consistently improves the three baseline policies over the 9 tasks from both the Google Robot and WidowX robotic platforms. On average, PCD boosts the success rates of the autoregressive policy `OpenVLA`, and the diffusion-based policies `Octo` and $\pi_0$ by up to **50.6%**, **29.7%** and **8.9%**, respectively. **2)** Our PCD exhibits stable and outstanding performance across the three object annotation strategies, including artificial Point and Box prompts and automatic object detection with Grounding DINO, underscoring its effectiveness and adaptability. In particular, on the `Octo` and `OpenVLA` baseline policies, leveraging the off-the-shelf model Grounding DINO demonstrates better performance than using human Point and Box prompts, while showing comparable results on $\pi_0$, highlighting the practical applicability of the PCD method. **3)** Compared to `OpenVLA` and `Octo`, the baseline model $\pi_0$ demonstrates significantly superior performance across all 9 tasks. For instance, `OpenVLA` and `Octo` achieve near-zero success rates on "Apple Drawer" task whereas $\pi_0$ achieves success rates of **17.0%** and **68.7%** on these two tasks, respectively. Despite this, from the average results over the 9 tasks, our PCD still enhances the strong baseline $\pi_0$ by **8.9%**. In summary, the obtained results in the table demonstrate that our PCD approach can be used as a plugin to improve different types of robot policies.

## 4.2 REAL-WORLD EXPERIMENTS

In this section, we conduct experiments on real-world tasks to assess PCD's practical effectiveness.

**Experimental Setup**. The simulation experiments show that $\pi_0$ significantly outperforms `Octo` and `OpenVLA`. Therefore, we employ $\pi_0$ as the baseline policy to validate the effectiveness of our PCD approach, as shown in Fig. 3. Considering the generalization limitations of existing robot policies in real-world environments, we first finetune $\pi_0$ on downstream real-world tasks before evaluating PCD's performance. We design 6 manipulation tasks that focus on evaluating the policy's performance along multiple dimensions: spatial reasoning and interacting with various objects and scenes. The $\pi_0$ policy is jointly fine-tuned on the 6 real-world tasks, with each task consisting of 10 training demonstrations. During inference, we conduct 20 trials for each real-world task and randomize the configurations and orientations of task-specific objects for each trial. We use an AGILEX PIPER 6DOF robot arm in our experiments. For our PCD, we leverage the off-the-shelf model Grounding DINO to detect task-specific objects in the initial observation, and use LaMa (Suvorov et al., 2022) to inpaint those objects segmented by the SAM2 (Kirillov et al., 2023) model. All other hyperparameters are kept consistent with the settings applied in the simulation experiments.

**Results and Analysis**. Fig. 3 reports the success rates as well as time costs of the state-of-the-art robot policy $\pi_0$ (Black et al., 2024) integrated w/ or w/o our PCD on 6 real-world tasks. For each task, we record the time cost (in seconds) from the start of the inference process to the successful completion of the task. From the obtained results in the figure, we have several key observations. **1)** PCD consistently improves the success rates of the strong baseline policy across six real-world tasks, achieving an average enhancement of **108%**. **2)** While PCD significantly boosts success rates, it also

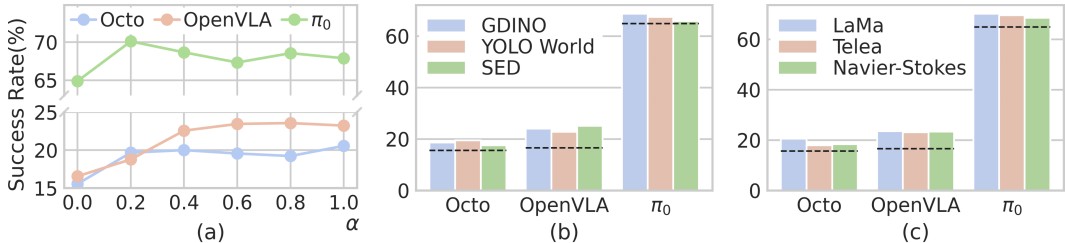

Figure 4: Ablation studies on **(a)** the hyperparameter $\alpha$ in Eq. (2), **(b)** the object detection schemes and **(c)** object inpainting strategies in Track2Mask. $\alpha = 0$ in **(a)** and the black dotted lines in **(b)(c)** represent the performance of the baseline policies. The results are averaged over the 9 simulation tasks. **PCD consistently improves the three policies when $\alpha > 0$ and exhibits low sensitivity to changes in off-the-shelf object detection and inpainting strategies.**

increases the time cost by 24% on average compared to the baseline policy. However, considering the notable improvement in task completion success rates, this trade-off is acceptable in a wide range of real-world robotic applications. **3)** PCD demonstrates consistent performance improvements across tasks of varying complexity. For example, in the challenging "Stack Cube" task, PCD improves the success rate from 0.05 to 0.10, showing its strong adaptability and robustness across diverse scenarios. Additionally, the backgrounds of these six real-world tasks are far more complex than those of the nine simulation tasks, containing numerous distracting elements such as plastic bottles, baskets, and trash cans. Nevertheless, our PCD shows remarkable advantages. It is worth mentioning that we use consistent PCD hyperparameters across tasks in both the simulation and real-world environments, without carefully tuning task-specific PCD hyperparameters (e.g., $\alpha$) for each individual task, leaving room for potential performance improvements through task-specific parameter tuning.

## 4.3 Ablation Studies

In this section, we conduct ablation studies to explore how the performance of our PCD varies with different design decisions. We perform experiments in the SIMPLER simulation environment, and report the average results across the nine tasks for our PCD applied to the three baseline policies.

**Effect of $\alpha$.** As formulated in Eq. (2), PCD leverages the hyperparameter $\alpha$ to control the level of amplification between output distributions from original and object-masked visual inputs. A larger $\alpha$ value indicates a stronger amplification of differences between the two distributions, while $\alpha = 0$ reduces to regular prediction. We adjust $\alpha$ by setting it to {0, 0.2, 0.4, 0.6, 0.8, 1.0}, and report the performance of our PCD method over the 9 tasks in Fig. 4 **(a)**. $\alpha = 0$ represents the baseline policies. As can be observed, our PCD consistently improves `Octo`, `OpenVLA` and $\pi_0$ when $\alpha > 0$, demonstrating its effectiveness and stability. Particularly, PCD yields the best results on `Octo`, `OpenVLA` and $\pi_0$ when the values of $\alpha$ are set to 1.0, 0.8, and 0.2, respectively.

**Effect of Off-the-shelf Object Detection Models**. Beyond leveraging artificial Point and Box prompts, the devised Track2Mask module can employ off-the-shelf open vocabulary object detection models to annotate task-specific objects in the initial observation of the trajectory during inference. Fig. 4 **(b)** presents an ablation study on various automatic object detection models, including the Grounding DINO (Liu et al., 2024b) model discussed in the main paper, as well as the open vocabulary object detection model YOLO World (Cheng et al., 2024) and semantic segmentation model SED (Xie et al., 2024). The output of these models is fed into the SAM2 (Kirillov et al., 2023) model for tracking and segmenting the detected objects across subsequent observations in the trajectory. As can be seen, our PCD consistently improves the baseline policies when integrated with each of the three off-the-shelf models, showcasing its effectiveness and stability. Generally, the Grounding DINO model yields the best results among the three models.

**Effect of Object Inpainting Strategies**. Once the objects specified in the language instruction of the target task are segmented by the SAM2 model, they are inpainted to create object-masked observations. In this experiment, we investigate the effect of different object inpainting strategies on PCD's performance. Fig. 4 **(c)** shows the results of PCD integrated with three inpainting strategies: Telea (Telea, 2004) and Navier-Stokes (Bertalmio et al., 2001) and LaMa (Suvorov et al., 2022). As indicated in the figure, our PCD approach consistently improves the baseline policies across all

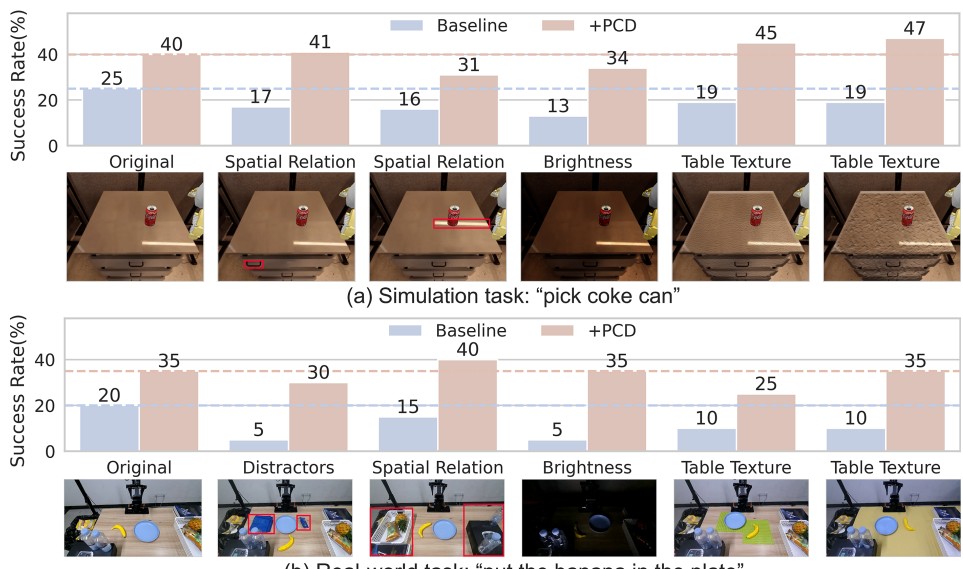

Figure 5: Performance of baseline policies integrated w/ or w/o our proposed PCD approach in unseen testing scenarios. Dotted lines represent the original performance. OpenVLA (Kim et al., 2024) and $\pi_0$ (Black et al., 2024) are used as baselines in simulation and real-world environments, respectively. **PCD effectively mitigates the side effects of various types of spurious correlations, boosting the robot policy's generalization to novel scenarios.**

strategies, showcasing its effectiveness and stability. Furthermore, PCD demonstrates low sensitivity to variations in the three object inpainting strategies, further validating its applicability and robustness. Notably, LaMa delivers the best results across all three baseline policies.

## 4.4 ROBUSTNESS TO VARIOUS KINDS OF SPURIOUS CORRELATIONS

The central concept of our proposed PCD approach in this work is to overcome the adverse effects of spurious features on the decision-making of robot policies. So, what types of spurious correlations can PCD effectively address? To answer the question, we evaluate the robustness of the OpenVLA (Kim et al., 2024) and $\pi_0$ (Black et al., 2024) policies to task scene variations in both simulation and real-world environments. As illustrated in Fig. 5, we report the performance of the two baseline policies integrated w/ or w/o our proposed PCD approach in different testing scenarios with unseen spatial relationships, brightness, distractors and table textures within visual observations. The results shown in the figure lead to the following observations. **1)** Consistent with the previous experimental results, our PCD approach significantly improves the performance of the baseline policies on both simulation and real-world tasks (Orange line vs. Blue line). **2)** Both baseline policies experience substantial drops in performance when testing scenarios are modified. For example, altering the brightness of visual observations results in **48**% and **75**% decreases in success rates for the two policies, respectively. **3)** PCD consistently alleviates the performance degradation of the baseline policies across all unseen testing scenarios. Remarkably, after integrating PCD, the two baseline policies even performs better in **4/10** of unseen scenarios than in the original training scenarios. The results in the figure reveal that our proposed PCD approach can serve as a plugin to mitigate the adverse effects of various types of spurious correlations on the generalization of robot policies.

## 5 CONCLUSION, LIMITATIONS AND FUTURE WORK

In this work, propose Policy Contrastive Decoding (PCD) to tackle the adverse effects of spurious correlations learned by generalist robot policies. PCD redirects the policy's attention from spurious features to object-relevant ones during inference by contrasting action probability distributions obtained from original and object-masked inputs. As a *plug-and-play* approach, PCD can enhance both autoregressive and diffusion-based policies, without requiring fine-tuning or access to pre-

trained model weights. Comprehensive experiments demonstrate PCD's flexibility and effectiveness, achieving consistent improvements over three state-of-the-art robot policies across a total of 15 tasks in simulation and real-world environments.

While our PCD demonstrates effectiveness and flexibility, there are certain limitations. **1)** PCD performs object detection and segmentation for each visual observation in the trajectory to produce object-masked observations, which increases the computational overhead of the baseline police. Recent progress in fast LLM inference techniques (Zhou et al., 2024; Leviathan et al., 2023; Liu et al., 2023c) could potentially enhance the computational efficiency of our PCD approach. **2)** This work concentrates exclusively on addressing the spurious correlation issue in robot policies during the testing stage, without exploring methods to prevent the learning of such correlations during training. Hence, future work will focus on investigating effective strategies to tackle spurious correlations at both the training and inference levels.

## ACKNOWLEDGEMENT

This work is supported by grants from the National Natural Science Foundation of China (No.62506310, No.62425208, No.U22A2097, No.U23A20315, No.82441006), the China Post-doctoral Science Foundation (No.2025M781517), and the Sichuan Science and Technology Program (No.2026NSFSC1479).

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

## A CAM VISUALIZATION OF FAILURE CASES

We present the CAM (Selvaraju et al., 2017) visualization of failure cases for the `OpenVLA`[1] model across the nine tasks in Fig. 6. For each task, the averaged CAM results over the seven action dimensions ($[\Delta x, \Delta y, \Delta z, \text{rot}_x, \text{rot}_y, \text{rot}_z, \text{gripper}]$) are plotted. As can be observed from the figure, the model tends to predict actions based on spurious or background features in visual observations. For instance, on the "move the blue plastic bottle near sponge" (denoted as "Move Near") task, the model model habitually focuses on the "coke can" object that it frequently encountered during the training phase. These observations demonstrate that the learned policy utilizes perceptual data for decision-making in a way that significantly diverges from human strategies. Robot policies often erroneously associate task-irrelevant features with actions, a major cause of overfitting and poor generalization across tasks.

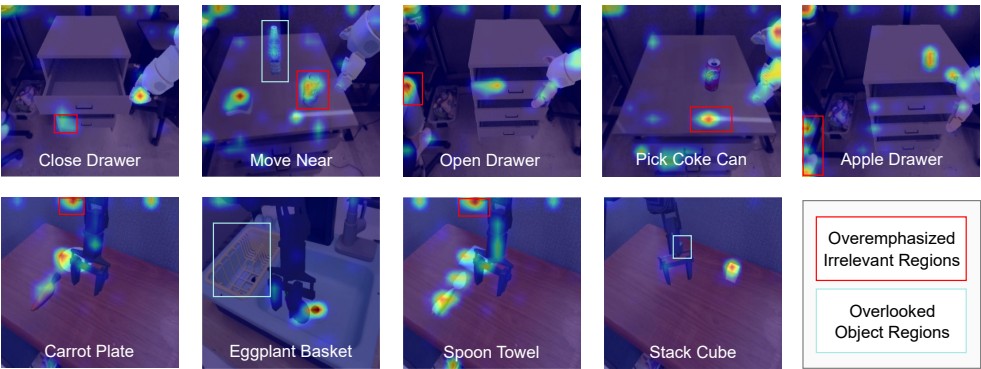

Figure 6: CAM visualization of failure cases.

## B DETAILS OF THE TRACK2MASK MODULE

In our devised Track2Mask module, the target object specified by the language instruction in the initial observation can be annotated using Point and Box prompts, along with off-the-shelf open-vocabulary object detection models, as illustrated in Fig. 7. The SAM2 (Kirillov et al., 2023) model is then applied to automatically track and segment the target object across subsequent observations in the trajectory, which enables precise object masking with minimal or no human intervention during inference. Note that, once the task-specific objects are segmented by the SAM2 model, they are inpainted to create object-masked observations.

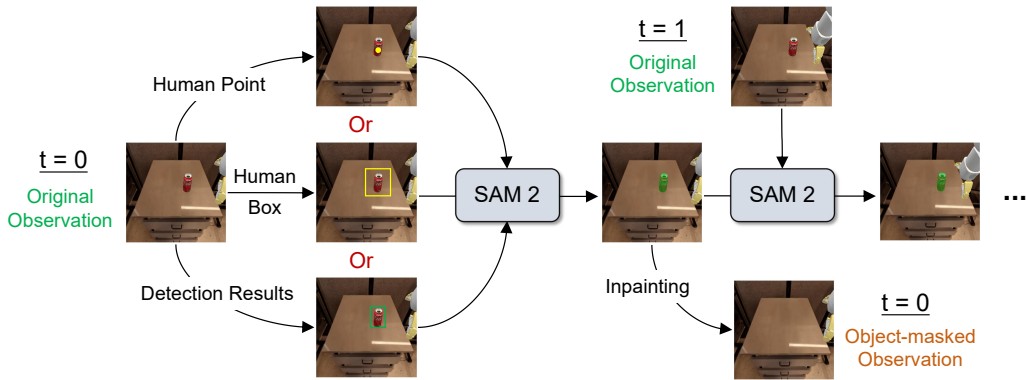

Figure 7: Illustration of the Track2Mask module.

---

[1]We only visualize the CAM results of `OpenVLA`, as the action prediction mechanism of diffusion-based models (e.g., `Octo` and $\pi_0$) makes it difficult to produce their CAM results.

## C  DETAILS OF THE EXPERIMENTAL SETUP

### C.1  BASELINE POLICIES

**Simulation Experiments**. We conduct simulation experiments using three diverse robot policies in the SIMPLER environment, including the autoregressive policy `OpenVLA` (Kim et al., 2024), and diffusion-based policies `Octo` (base) (M. et al., 2024) and $\pi_0$ (Black et al., 2024).

- `OpenVLA`: a vision-language-action model with 7 billion parameters, is trained on 970,000 episodes of robotic demonstrations from the Open X-Embodiment dataset. This policy is fine-tuned using the pre-trained Prismatic (Karamcheti et al., 2024b) model.[2]

- `Octo` (base): an open-source generalist policy with 93 million parameters, is pre-trained on a blend of 25 different datasets from the Open X-Embodiment dataset. It utilizes a transformer backbone derived from ViT-B, accompanied by a diffusion action head to model expressive action distributions.[3]

- $\pi_0$: a flow matching architecture built on top of a pre-trained vision-language model to leverage Internet-scale semantic knowledge. To facilitate evaluation in the SIMPLER environment, we leverage the reproduced version of $\pi_0$, pretrained on tasks from this environment.[4]

**Real-world Experiments**. Since the simulation experiments show that $\pi_0$ significantly outperforms `Octo` and `OpenVLA`, we conduct real-world experiments using the $\pi_0$ policy [5]. The policy is finetuned using on six real-world tasks, with each task consisting of 10 trajectories. We employ the parameter-efficient finetuning strategy LoRA (Hu et al., 2022) to adapt $\pi_0$ to these real world tasks. The training hyperparameters are listed in Table 2.

Table 2: Hyperparametes for finetuning $\pi_0$ on real-world tasks.

| Hyperparametes | Setting |
|---|---|
| Batch Size | 128 |
| Optimizer | AdamW |
| Learning Rate | 2.5e-5 |
| LR Schedule | Cosine Decay |
| Weight Decay | 1e-10 |
| Training Step | 25000 |
| Action Chunk | 10 |
| LoRA Rank | 16 |

### C.2  EVALUATION TASKS

We conduct extensive evaluations in both simulation and real-world environments using a total of 15 diverse tasks. The details of these tasks are presented in Table 3. We conduct 20 trials for each real-world task, randomizing the task configurations in every trial, as illustrated in Fig. 8.

## D  PERFORMANCE AT THE MAXIMUM STEP

In Table 1, tasks completed before the predefined maximum step are included in the success rate calculation, and the task is terminated upon completion. Table 4 presents success rates computed at the maximum step, irrespective of whether tasks were completed earlier. As seen, PCD demonstrates its advantages by improving the success rates of `OpenVLA`, `Octo` and $\pi_0$ by up to **45.0%**, **19.3%** and **7.9%**, respectively. We also observe that the vast majority of models underperform on these tasks relative to their results in Table 1, which indicates that even though they complete the task successfully ahead of the maximum steps, their later predicted actions lead to task failure. The

---

[2] `https://openvla.github.io/`

[3] `https://octo-models.github.io/`

[4] `https://github.com/allenzren/open-pi-zero`

[5] `https://www.physicalintelligence.company/blog/pi0`

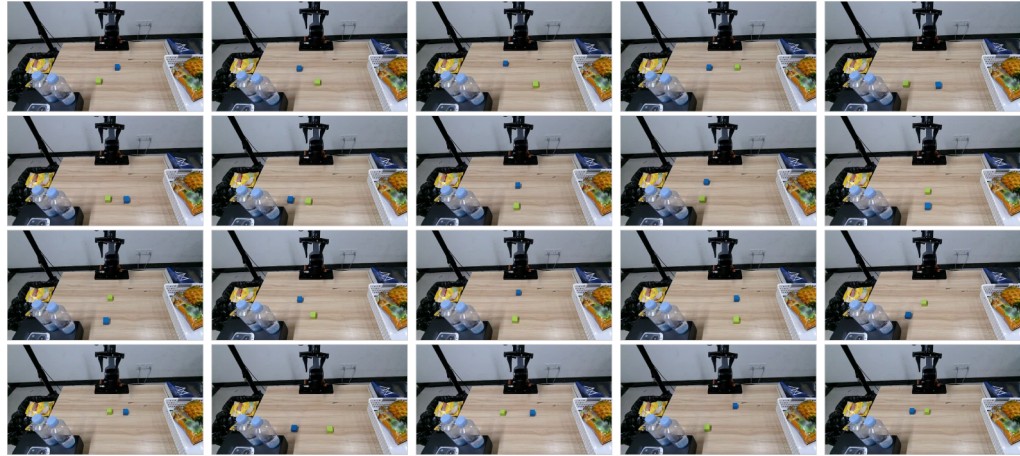

Figure 8: We conduct 20 trials for each real-world task, randomizing the task configurations in every trial. The language command for this task is "stack the green cube on the yellow cube."

Table 3: Details of simulation and real-world tasks.

| Env. | Tasks | Language Instruction | Task-specific Objects | # Trials |
|---|---|---|---|---|
| Simulation | Close Drawer | close top/middle/bottom drawer | top/middle/bottom drawer | 300 |
| | Move Near | move X near Y | apple, orange, pepsi can, ... | 300 |
| | Open Drawer | open top/middle/bottom drawer | top/middle/bottom drawer | 300 |
| | Pick Coke Can | pick coke can | coke can | 300 |
| | Apple Drawer | open top drawer/place apple into top drawer | top drawer, apple | 300 |
| | Carrot Plate | put carrot on plate | carrot, plate | 300 |
| | Eggplant Basket | put eggplant into yellow basket | eggplant, yellow basket | 300 |
| | Spoon Towel | put the spoon on the towel | spoon, towel | 300 |
| | Stack Cube | stack the green block on the yellow block | green cube, yellow cube | 300 |
| Real-world | Pick Ball | pick the ball | ball | 20 |
| | Pick Plug | pick the plug | plug | 20 |
| | Move Near | move the can near the apple/watermelon | can, apple, watermelon | 20 |
| | Cookies Towel | put the cookies on the towel | cookies, towel | 20 |
| | Banana Plate | put the banana in the plate | banana, plate | 20 |
| | Stack Cube | stack the green cube on the yellow cube | green cube, yellow cube | 20 |

Table 4: Success rates (%) and 95% confidence interval of three baseline policies integrated w/ or w/o our PCD over 9 tasks from Google Robot and WidowX—**calculated at the maximum step**.

| | Model | OpenVLA(Kim et al., 2024) | | | | Octo (M. et al., 2024) | | | | $\pi_0$ (Black et al., 2024) | | | |
|---|---|---|---|---|---|---|---|---|---|---|---|---|---|
| | | Base | **+PCD** | | | Base | **+PCD** | | | Base | **+PCD** | | |
| | | | Point | Box | DINO | | Point | Box | DINO | | Point | Box | DINO |
| | **Average** | 15.1±1.3 | 21.0±1.5 | 21.1±1.5 | **21.9**±1.6 | 11.9±1.2 | 13.4±1.3 | 14.0±1.3 | **14.2**±1.3 | 59.2±1.9 | 62.7±1.8 | 63.7±1.8 | **63.9**±1.8 |
| | | | +39.1% | +39.7% | +45.0% | | +12.6% | +17.6% | +19.3% | | +5.9% | +7.6% | +7.9% |
| Google Robot | Close Drawer | 47.3±5.6 | 61.3±5.5 | 63.7±5.4 | **72.7**±5.0 | 28.0±5.1 | 24.3±4.9 | **28.3**±5.1 | 20.3±4.6 | 75.0±4.9 | 74.0±5.0 | **79.0**±4.6 | 74.7±4.9 |
| | Move Near | 47.3±5.6 | 48.0±5.7 | **50.7**±5.7 | 41.3±5.6 | 2.7±1.8 | 5.0±2.5 | 4.7±2.4 | **8.0**±3.1 | 60.0±5.5 | 56.0±5.6 | 59.0±5.6 | **62.0**±5.5 |
| | Open Drawer | 23.3±4.8 | **34.3**±5.4 | 31.0±5.2 | 33.3±5.3 | 0.3±0.7 | 0.3±0.7 | 0.0±0.0 | **1.0**±1.1 | 34.7±5.4 | 42.3±5.6 | 47.7±5.7 | **52.7**±5.6 |
| | Pick Coke | 17.3±4.3 | 34.0±5.4 | **41.3**±5.6 | 39.7±5.5 | 23.0±4.8 | 41.7±5.6 | 42.7±5.6 | **43.3**±5.6 | 80.3±4.5 | 81.0±4.4 | 82.3±4.3 | **84.0**±4.1 |
| | Apple Drawer | 0.0±0.0 | 0.3±0.7 | **1.0**±1.1 | 0.7±0.9 | **0.0**±0.0 | **0.0**±0.0 | **0.0**±0.0 | **0.0**±0.0 | 17.0±4.3 | 26.0±5.0 | 26.7±5.0 | **27.3**±5.0 |
| WidX. | Carrot Plate | 0.0±0.0 | **3.0**±1.9 | 0.0±0.0 | 2.7±1.8 | 9.0±3.2 | **12.7**±3.8 | 11.7±3.6 | 12.0±3.7 | 54.0±5.6 | **62.0**±5.5 | 55.3±5.6 | 54.3±5.6 |
| | Eggplant Basket | 0.3±0.7 | **5.0**±2.5 | 1.7±1.4 | 3.7±2.1 | **37.0**±5.5 | 25.3±4.9 | 22.7±4.7 | 31.7±5.3 | **83.3**±4.2 | 77.7±4.7 | 80.3±4.5 | 81.0±4.4 |
| | Spoon Towel | 0.0±0.0 | 1.7±1.4 | 0.3±0.7 | **3.0**±1.9 | 7.0±2.9 | 9.0±3.2 | **13.7**±3.9 | 9.0±3.2 | 79.3±4.6 | **85.0**±4.0 | 82.7±4.3 | 82.7±4.3 |
| | Stack Cube | 0.0±0.0 | **1.3**±1.3 | 0.0±0.0 | 0.0±0.0 | 0.0±0.0 | **2.7**±1.8 | 2.3±1.7 | 2.3±1.7 | 49.0±5.7 | **60.3**±5.5 | **60.3**±5.5 | 56.7±5.6 |

superiority of the proposed PCD method is compellingly evidenced by the consistent performance gains over baseline policies presented in Table 1 and Table 4.

# E    COMPUTATIONAL OVERHEAD

Table 5 reports the computational overhead of three baseline policies integrated w/ or w/o our PCD method in the SIMPLER environment. **1) Inference Latency**. PCD approximately doubles the

Table 5: Computational overhead of three baseline policies integrated w/ or w/o our PCD method in the SIMPLER environment.

| Model | OpenVLA | | Octo | | $\pi_0$ | |
|---|---|---|---|---|---|---|
| | Base | +PCD | Base | +PCD | Base | +PCD |
| Average time cost for each infer. step (s) | 0.86 | 1.77 | 0.21 | 0.39 | 0.66 | 1.09 |
| Memory cost (MB) | 16357 | 16869 | 2884 | 3528 | 8535 | 11699 |

inference latency per step across all three policies. This is an expected trade-off, as PCD requires a second forward pass on the object-masked input to compute the contrastive signal. For Octo and $\pi_0$, although the KDE-PM process introduces additional inference time, the overall time overhead is still kept to roughly a twofold increase due to their parallel processing of the original and object-masked images. Crucially, this increased inference time does not significantly affect the total task completion time in real-world scenarios, given the much longer duration of the robot's physical execution. As proven by Figure 3, PCD only brings 24% extra execution time on OpenVLA. **2) Memory Cost**. The impact on memory cost varies based on the policy's architecture. For OpenVLA, which processes the original and masked inputs serially, the memory overhead is negligible, with the primary cost being the increased sequential processing time. In contrast, the memory costs of Octo and $\pi_0$ are $1.22\times$ and $1.37\times$ that of the baseline respectively, due to their parallel data processing mechanisms.

## F    PCD VS. CLASSIFIER-FREE GUIDANCE (CFG)

The proposed PCD algorithm shares similarity with the training-free approach classifier-free guidance (CFG) (Ho and Salimans, 2022), where the masked image serves as the unconditional inputs. However, the essential distinction lies in how and when the guidance is applied: CFG provides implicit guidance during the iterative decoding process. At each step of the diffusion, it steers the generation away from the unconditional prediction. This is an implicit probabilistic modeling process. In contrast, our PCD is an explicit post-hoc correction. The quantitative comparison shown in Table 6 shows that our PCD method significantly outperforms CFG across the nine SIMPLE tasks. However, the time cost of PCD is 1.29 times that of CFG. Therefore, harnessing the complementary strengths of PCD and CFG to further address spurious correlations remains a promising avenue for future research.

Table 6: Comparison of PCD and Classifier-Free Guidance (CFG) (Ho and Salimans, 2022).

| Model | $\pi_0$ | +CFG | **+PCD** |
|---|---|---|---|
| Average time cost for each inference step (seconds) | 0.66 | 0.84 | 1.09 |
| Average success rate over nine tasks (%) | 63.9 | 62.7 | **68.1** |
| Close Drawer | 75.7 | 75.7 | 74.7 |
| Move Near | 67.3 | 62.7 | 66.7 |
| Open Drawer | 38.0 | 46.3 | 47.7 |
| Pick Coke | 84.0 | 81.3 | 84.3 |
| Apple Drawer | 17.0 | 13.0 | 26.0 |
| Carrot Plate | 58.0 | 59.0 | 67.7 |
| Egg. Basket | 86.0 | 87.3 | 83.7 |
| Spoon Towel | 80.7 | 78.3 | 86.3 |
| Stack Cube | 68.7 | 60.3 | 76.3 |

To further investigate CFG, we conduct a step-by-step analysis in Table 7, applying CFG after a certain step and running 100 trials for each setting. The results show a clear trend: the earlier the CFG intervention, the more severe the performance degradation. The primary reason appears to be that CFG's influence on the noise space disrupts the semantic integrity of the model's reasoning process, ultimately degrading performance.

Table 7: Performance of CFG under different starting steps.

| Starting step | Success Rate (%) |
|---|---|
| 0 | 62.7 |
| 2 | 63.8 |
| 4 | 64.8 |
| 8 | 64.5 |

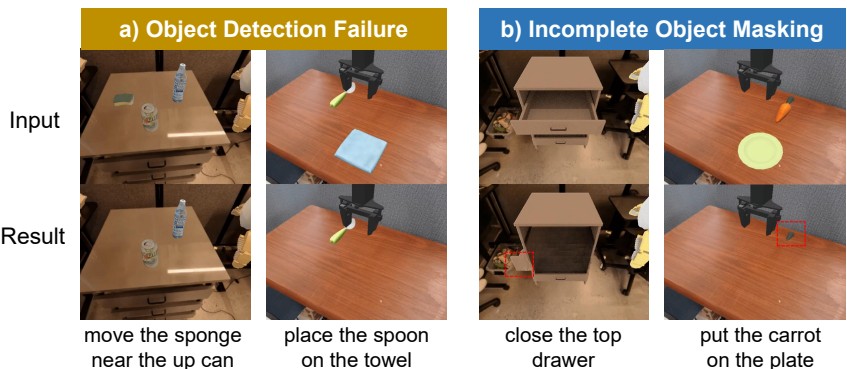

Figure 9: Failure cases of the Track2Mask module.

## G    FAILURE CASES OF TRACK2MASK

From the Track2Mask pipeline presented in Appendix Fig.7, the failure cases fall into two categories: **a) Object Detection Failure**—the off-the-shelf open-vocabulary detector (i.e., GDINO) fails to localize objects in the initial observation; **b) Incomplete Object Masking**—the target objects along the trajectory are partially masked, as shown in Fig. 9. According to Equation 2, when the first failure case occurs, PCD's prediction becomes equivalent to the original baseline result.

We conduct a ablation study to assess the impact of the incomplete masking failure cases in Table 8, where $\beta$ indicates the ratio of masked pixels manually excluded. As can be seen, PCD's performance progressively declines as $\beta$ increases, until it approaches the baseline performance when $\beta$ reaches 60%. Nevertheless, both kinds of failure cases are exceptionally rare in our experiments, owing to the stability and effectiveness of the employed off-the-shelf models, i.e., GDINO for object detection and SAM v2 for object tracking and segmentation.

Table 8: Performance of PCD under different ratios ($\beta$) of masked pixels manually excluded. The task is "pick coke can".

| Model | Baseline | +PCD ($\beta = 0$) | $\beta = 0.2$ | $\beta = 0.4$ | $\beta = 0.6$ |
|---|---|---|---|---|---|
| OpenVLA | 25 | 40 | 32 | 28 | 27 |
| $\pi_0$ | 84 | 88 | 88 | 85 | 84 |

## H    APPLICABILITY TO MORE COMPLEX TASKS

In our experiments on multi-objects tasks, PCD masks all task-relevant objects simultaneously. Here, we perform a new ablation study using the "Move Near" task to validate this strategy. Specifically, for task "Move A Near B", we compare the results of i) masking only object A, ii) masking only object B, iii) masking A first, followed by masking B upon a successful grasp of A; and iv) Masking both A and B. The results are shown in Table 9, where the baseline model is OpenVLA. As can be observed, masking all task specific objects simultaneously yields the best performance.

Table 9: Ablation study on masking strategies for multi-object tasks.

| Model | Task | Mask A | Mask B | Mask A then B | Mask A and B |
|---|---|---|---|---|---|
| OpenVLA | Move Near | 60 | 50 | 57 | 62 |
| $\pi_0$ | Carrot Plate | 53 | 62 | 56 | 67 |

Our PCD method has been evaluated on the long-horizon task "Apple Drawer" in the SIMPLER environment. As illustrated in Appendix Table 3, the official language instruction of the task is composed of two sequential sub-tasks: "open the top drawer" and "place the apple into the top drawer". Our strategy is to mask objects based on the instruction of the current sub-task. For sub-task 1, we mask only the "top drawer"; for sub-task 2, we mask both the "apple" and the "top drawer". In other words, PCD holds potential for long-horizon tasks by leveraging an LLM-based planner with CoT reasoning to decompose abstract commands into concrete sub-tasks.

## I PERFORMANCE GAPS AMONG DIFFERENT OBJECT ANNOTATION STRATEGIES

The observed performance gaps among different object annotation strategies in Table 1 arises from the distinct "preferences" inherent in each approach. For instance, prompts from humans versus those from object detection models exhibit different understandings of object presence. As illustrated in Fig.10, in the case of a "drawer", a human annotator tends to point to or box a specific drawer within a cabinet, whereas an object detection model typically bounds the entire cabinet containing all drawers. This leads to significant performance disparities on "drawer"-related tasks. We argue that no single object annotation strategy is universally optimal; rather, the most effective approach is contingent upon the specific task.

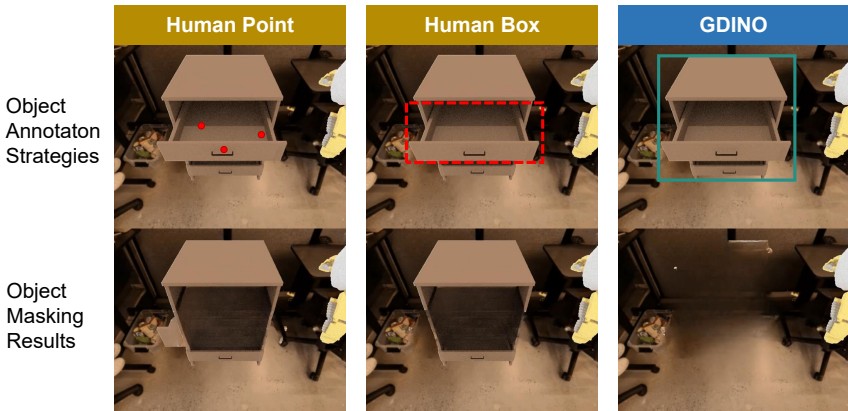

Figure 10: Object masking results of Track2Mask using different object annotation strategies.

## J ROBUSTNESS UNDER VARIOUS MAGNITUDES OF DISTRACTORS

Here, we investigate the robustness of the PCD method under various magnitudes of distractors by incrementally increasing the number of distractors in the image background of the "pick coke can" task. The obtained results confirm that PCD maintains superior performance and exhibits greater robustness against these disturbances compared to the baseline.

| Model | Original | Few distractors | Many distractors |
|---|---|---|---|
| OpenVLA | 25 | 24 | 14 |
| **+PCD** | 40 | 39 | 36 |

Table 10: Robustness of PCD to various magnitudes of distractors presented in Fig.11.

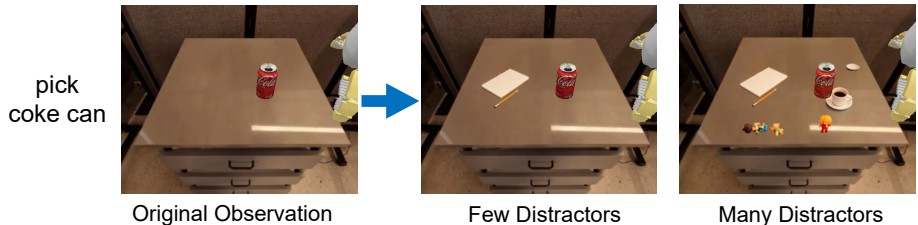

Figure 11: Various magnitudes of distractors.

## K PERFORMANCE IN MULTI-PERSPECTIVE SCENARIOS

In this part, we conduct an experiment to investigate PCD's effectiveness in multi-perspective scenarios. Specifically, we leverage the miniVLA+VQ h8+Wrist (Belkhale and Sadigh, 2024)

policy—which incorporates both third-person perspective and wrist images as input—as our baseline. The achieved results on five randomly sampled tasks from the LIBERO-90 (Liu et al., 2023a) benchmark are shown in Table 11, demonstrating that PCD consistently improves the baseline policy across all five tasks.

| LIBER-90 Task | miniVLA+VQ h8+Wrist | +PCD |
|---|---|---|
| put the butter at the front in the top drawer of the cabinet and close it | 54 | **72** |
| put the black bowl on top of the cabinet | 98 | **100** |
| pick up the ketchup and put it in the basket | 72 | **84** |
| put the white mug on the plate | 86 | **88** |
| pick up the book and place it in the front compartment of the caddy | 46 | **54** |
| Average | 71.2 | **79.6** |

Table 11: Performance of PCD on five LIBER-90 Tasks.

## L LLMs USAGE STATEMENT

In the preparation of this paper, we employed Large Language Models (LLMs) solely as a writing assistance tool for limited text polishing and language refinement. LLMs were not involved in any aspects of research ideation, conceptual development, technical analysis, algorithm design, experimental execution, or result interpretation. All scientific contributions, methodological innovations, and intellectual content remain entirely our own.

