# OpenReview forum: "Policy Contrastive Decoding for Robotic Foundation Models"
_ICLR.cc/2026/Conference — ICLR 2026 Poster_

### Official Review · Reviewer_gdDA · 2025-10-26

**Soundness:** 3
**Presentation:** 3
**Contribution:** 2
**Rating:** 4
**Confidence:** 4

**Summary:**

This paper proposes Policy Contrastive Decoding (PCD), a training-free plugin for pretrained robotic foundation models. It aims to redirect pre-trained robotic foundation models away from spurious visual correlations toward object-relevant cues. The key idea is: given a visual observation and language instruction, the robot policy is run twice on the original image and on a version where task-relevant objects are masked. It can be applied to both autoregressive and diffusion-based policies, showing its versatility.

**Strengths:**

1. The method does not need retraining or fine-tuning of a large robot policy, showing practicality in robotics, as VLA training is expensive.

2. The problem that the paper tries to solve is important: reliance on spurious visual correlations leading to poor generalization when distribution shifts happen.

3. The authors evaluate across multiple policies (i.e., autoregressive and diffusion-based) and in both simulation and real-world robot settings. They also include ablations, which help to analyze design choices.

**Weaknesses:**

1. It lacks some analysis for hyperparameters.

2. It does not include training and implementation details.

3. The paper defines “spurious correlations” formally (Definition 3.1) but then moves to a generic contrastive decoding mechanism without carefully clarifying which types of spurious correlations are mitigated and which are not. For instance, masking the object may remove task-relevant cues if the object is part of the visual context, yet the method tacitly assumes that the object is the only relevant cue and everything else is spurious. This assumption is too strong for many robotics settings.

4. The ablations, while present, focus mainly on hyperparameter $\alpha$, object detection model choice, and inpainting method. There is little ablation on the magnitude of spurious correlation removal, or how performance degrades when masking fails.

**Questions:**

1. What is the value of $\alpha$ in the main simulation experiment? Is it fixed or adapted to the VLAs?

2. For diffusion-based policies with KDE-PM, why do the authors choose 24 for experiments? Do authors have any time/memory trade-offs for this method compared to standard inference?

3. When the PCD is applied to VLAs, the success rate is improved but can not reach 100%, nor even 80% in most tasks. Why does it fail? Is it because PCD can not fully solve the spurious correlations problem?

4. How does the performance of PCD degrade when object detection or masking fails (e.g., false negatives, multiple objects, occlusion)? Can authors provide results showing robustness to mask quality or missing objects?

5. It seems that all the experiments were completed from a single perspective. How about multiple perspectives?

---

> ### Author Response · Authors · 2025-11-20
> **Response to Reviewer gdDA**
>
> We appreciate your valuable comments. We hope the following response can address your concerns.
>
> **1) It Lacks Some Analysis of Hyper-parameters**\
> Our ablation study on the key hyper-parameter $\alpha$ is presented in Section 4.3.  Following your suggestion, we performed a new ablation study on $N$, the number of sampled noise vectors in KDE-PM. The experimental results in the table below indicate that the performance improvement of PCD gradually stabilizes as $N$ increases to 24 (the baseline is $\pi_0$). Although continuing to increase $N$ to 32 can further boost performance, this would incur greater overhead for the KDE-PM process. Therefore, we set $N$=24.
> |N| Success Rate (%) |
> |---|----------------|
> |8| 67.1|
> |16|67.9|
> |24|69.1|
> |32|69.8|
>
> **2) It Does Not Include Training and Implementation Details**\
> Please refer to **Appendix C** for training and implementation details.
>
> **3) The Definition of “Spurious Correlations”**\
> The PCD framework itself is not restricted by this assumption. PCD is a flexible meta-algorithm designed to incorporate human priors on what constitutes a task-relevant versus a spurious feature. Our use of target object-masking via the Track2Mask module is merely an instance of this framework, leveraging the most common and immediate human prior in manipulation tasks. If the prior information suggested other cues were task-relevant—e.g., a specific background feature, the robot's own body, or other contextual objects (as in multi-object interaction tasks)—the Track2Mask strategy could be adapted to mask all those elements in the observation.
>
> In essence, we embrace the "no free lunch" principle: the extra performance boost from PCD fundamentally comes from integrating this explicit, human-defined prior. For tasks where the definition of task-relevant features extends beyond the target object, adapting the masking strategy is a necessary and flexible step within our framework. Exploring novel masking strategies beyond single-object removal is a promising direction for future work.
>
> **4) Illustration of the Magnitude of Spurious Correlation Removal**\
> As PCD modifies the policy's final predictions without influencing the intrinsic decoding process, directly visualizing the magnitude of spurious correlation removal is infeasible.
> We present an indirect demonstration of PCD’s effectiveness against varying magnitudes of spurious correlation in **Appendix J**. Specifically, we incrementally increased the background perturbation of the original "pick coke can" task by adding of distractors. The results in the table below confirm that PCD maintains superior performance and exhibits greater robustness against these disturbances compared to the baseline.
> |Model|Origina Observationl|w/ Few Distractors|w/ Many Distractors|
> |-----| ----| ------ |---- |
> |OpenVLA|25|24|14|
> |**+PCD**|40|39|36|
>
> **5) Why the Success Rate Can Not Reach 100% When Applying PCD to Baselines**\
> Our PCD functions as a refinement module to correct a policy's existing behaviors, rather than to generate novel skills for tasks on which the base model completely fails. That is to say, if the model lacks the fundamental capability to perform the task, merely focusing its attention on correct target objects via PCD is insufficient to ensure completion. In addition to spurious correlations, task execution success is influenced by numerous other factors, such as task hardness and robot's physical state.
>
> **6) Performance in Multi-perspective Scenarios**\
> Following your suggestion, we conducted a new experiment to investigate PCD’s effectiveness in multi-perspective scenarios. Specifically, we leveraged the miniVLA+VQ h8+Wrist [1] policy—which incorporates both third-person perspective and wrist images as input—as our baseline. The achieved results on five randomly sampled tasks from LIBERO-90 [2] are shown in the table below, demonstrating that PCD consistently improves the baseline policy across all five tasks.
> |LIBERO-90 Task|miniVLA+VQ h8+Wrist|+PCD|
> | -------------------------------------------------------------------------| ------------ | ------- |
> |put the butter at the front in the top drawer of the cabinet and close it | 54| **72**|
> |put the black bowl on top of the cabinet|98|**100**|
> |pick up the ketchup and put it in the basket |72| **84**|
> |put the white mug on the plate|86|**88**|
> |pick up the book and place it in the front compartment of the caddy|46|**54**|
> |Average|71.2| **79.6**|
>
> Additionally, our open-source implementation offers well-designed interfaces that enable users to easily switch to multi-perspective scenarios. For details, please refer to: https://github.com/pcd-robot/PCD-real (/src/openpi/policies/policy.py).
>
> [1] S. Belkhale et al. Minivla: A better vla with a smaller footprint. https://ai.stanford.edu/blog/minivla/, 2024\
> [2] B. Liu, et al. LIBERO: Benchmarking knowledge transfer for lifelong robot learning. NeurIPS 2023.
>
> We have add this discussion to **Appendix  K** of the revised paper.

---

> > ### Comment · Reviewer_gdDA · 2025-11-24
> >
> > Thank you for your response and the additional experiments. However, I still have the following concerns:
> >
> > 1. Regarding Tracking-to-Mask, it is mentioned on line 224 that it requires minimal or no human intervention. Following the paragraph, I believe there is no human intervention in this phase. Can you explain when you need human intervention? Is human intervention important in your method?
> >
> > 2. Does Task2Target sometimes fail in your experiments? If so, when does it fail?
> >
> > 3. Can you explain some of the failure cases of the method, like OpenVLA + PCD Box in WidowX Stack Cube?
> >
> > I am willing to raise my score if the authors can keep discussing with me and provide a convincing explanation.

---

> ### Author Response · Authors · 2025-11-24
> **Response to Reviewer gdDA**
>
> We appreciate your valuable feedback. We hope the following response can address your concerns.
>
> **1) When PCD Needs Human Intervention?**\
> The phrase "minimal or no human intervention" refers to the two flexible initialization modes of our Track2Mask module:
> **a) Automatic Mode w/o Human Intervention.** We employ off-the-shelf open-vocabulary detectors, such as Grounding DINO (GDINO), to automatically detect target objects in the initial observation based on object names derived from the language instruction. In this mode, the entire pipeline is fully automated. **b) Interactive Mode w/ Minimal Human Intervention.** We also provide the option to annotate task-specific objects via Point or Box prompts, which requires human intervention solely for the initial observation. Once annotated, the SAM 2 model automatically tracks and masks these objects throughout the rest of the trajectory, with no further human intervention.
>
> Please refer to **Appendix B (Fig. 7)** for more details of our devised Track2Mask module.
>
> **2) Failure Cases of the Tracking2Mask Module?**\
> The failure cases of Track2Mask fall into two categories: **a) Object Detection Failure**—the off-the-shelf open-vocabulary detector (i.e., GDINO) fails to localize objects in the initial observation. **b) Incomplete Object Masking**—the target objects along the trajectory are partially masked.
>
> Please refer to our **Response to Common Questions-Part II** or **Appendix G** of the revised paper for more details.
>
> **3) Explanation of PCD’s Failure Cases**\
> As you noted, PCD led to a performance decrease for OpenVLA on the "Stack Cube" task. We think the key reasons are twofold. **a) Trajectory Overfitting.** We closely examined OpenVLA's execution trajectories on this WidowX task and observed a high degree of trajectory overfitting, characterized by a fixed downward motion. We observed that even slight perturbations to the cube's position caused the success rate to zero, revealing that OpenVLA fundamentally lacks the capability to execute this task. This aligns with existing findings [1, 2] indicating that OpenVLA significantly underperforms on WidowX tasks compared to other models—though the underlying cause remains an open question. As we emphasized, our PCD functions as a refinement module to correct a policy's existing behaviors, rather than to generate novel skills for tasks on which the base model completely fails. **b) Unified Hyper-parameters.** To demonstrate generalizability, we uniformed hyper-parameter setting (e.g., $\alpha$) across all tasks. However, optimal parameters vary by task and baseline model. To demosntrate this, we report the results of Octo+PCD (w/ Point annotation, 300 trials) on the "Close Drawer" task in the following table. As indicated, the optimal $\alpha$ value for this task on Octo is 0.2, while a value of 1.0 was used in our experiments. Therefore, we believe that adopting adaptive or task-specific tuning in future work would address this limitation and improve performance on sensitive tasks.
>  | Task    | Octo | **+PCD** | $\alpha$=0.2       | $\alpha$=0.4 | $\alpha$=0.6 | $\alpha$=0.8 | $\alpha$=1.0 (used in our experiments) |
>  | ------------ | -------- | -------- | ------------------ | --- | --- | --- | --- |
>  | Close Drawer | 31.0 |      | **42.3** | 35.7  | 36.3  | 34.3  | 26.0  |
>
> Additionally, we also observe from Table 1 that PCD fails to improve the Octo model on the "Apple Drawer" task, where the success rate remains 0%. This further highlights a fundamental characteristic of our method: PCD functions as a refinement module to correct a policy's existing behaviors, rather than to generate novel skills for tasks on which the base model completely fails. In other words, if the baseline model’s initial output distribution assigns a near-zero probability to the correct action, PCD lacks a valid signal to amplify, resulting in minimal improvement.
>
> [1] https://github.com/simpler-env/SimplerEnv/issues/92 \
> [2] https://github.com/simpler-env/SimplerEnv/issues/78
>
>
> We remain available and would be happy to address any further questions.

---

> > ### Comment · Reviewer_gdDA · 2025-11-24
> >
> > Thank you for the detailed reply. Based on the experimental results, I think your method can achieve the best performance and full automation without any human intervention, which is something you should highlight. In addition, I still have some concerns about hyperparameter tuning. Nevertheless, the results show that most values of alpha can still bring some improvement, which reduces the effort needed for tuning. Therefore, I will increase my score to 6.

---

> > > ### Author Response · Authors · 2025-11-25
> > > **Thanks**
> > >
> > > Thank you for your active feedback and engagement during the rebuttal process. We appreciate your endorsement.

---

### Official Review · Reviewer_DdfD · 2025-10-29

**Soundness:** 4
**Presentation:** 4
**Contribution:** 4
**Rating:** 8
**Confidence:** 4

**Summary:**

This paper proposes "Policy Contrastive Decoding" (PCD), a novel method to address the issue of "spurious correlations" in robotic foundation models, where policies incorrectly learn from task-irrelevant features like background or lighting instead of the target objects. PCD is a "training-free" and "plug-and-play" approach that works by contrasting the action probability distribution from the original visual input with a distribution derived from an "object-masked" version of the same input. This process, which requires no model retraining, amplifies the policy's focus on object-relevant features. The method is designed to be flexible, using a "Tracking-to-Mask" (Track2Mask) strategy to identify and mask objects automatically, and a "KDE-based Probabilistic Modeling" (KDE-PM) scheme to make it compatible with both autoregressive (like OpenVLA) and diffusion-based (like Octo and $\pi_0$) policies. Experiments in simulation and the real world show that PCD significantly improves the success rates of these state-of-the-art policies, enhancing the $\pi_0$ policy by 8.9% in simulation and 108% in the real world.

**Strengths:**

- The paper identifies and addresses a critical, known weakness in robotic foundation models: their tendency to learn "spurious correlations" (e.g., relying on background or lighting) rather than the actual task-relevant objects, which hurts generalization.

- The proposed "Policy Contrastive Decoding" (PCD) is a major strength because it is a "plug-and-play" solution. It can be applied to existing, pre-trained policies without requiring any costly retraining, fine-tuning, or access to the model's internal weights.

- The method is not designed for just one type of model. It introduces a "KDE-based Probabilistic Modeling" (KDE-PM) scheme that allows it to work with *both* autoregressive policies (like OpenVLA) and diffusion-based policies (like Octo and $\pi_0$).

- PCD demonstrates significant performance gains on top of three different state-of-the-art policies. It achieves impressive improvements, boosting the strong $\pi_0$ baseline by 8.9% in simulation and, most notably, by 108% in real-world tasks.

- The paper includes extensive experiments to prove its claims, and it is open-sourced.

**Weaknesses:**

- The method adds significant computational overhead and latency at inference time. The paper notes it "approximately doubles the inference latency per step" and adds a 24% total time cost to real-world tasks. This may influence the performance on highly dynamic tasks,

- The method introduces a new, highly sensitive hyperparameter, $\alpha$, that controls the strength of the correction. Ablation studies show that the best $\alpha$ is different for each model (e.g., 0.2 for $\pi_0$ but 1.0 for Octo), meaning it requires careful, model-specific tuning, which undermines the "plug-and-play" claim.

- The authors admit the method is a patch at inference time, not a fix. It only addresses the symptoms of spurious correlations during testing and does nothing to prevent the model from learning them during training in the first place.

- The real-world experiments, while showing a large relative gain (108%), were conducted on a $\pi_0$ model that was already fine-tuned on 10 demonstrations for each specific task. This makes it unclear how PCD would perform on a general-purpose model in a true zero-shot, real-world setting. Also, the performance gain of 108% in real-world compared to the 8.9% in the simulation is very large, which seems unusual.

- Even with PCD, the absolute success rates in many challenging real-world tasks remain low (e.g., "Stack Cube" improves from only 5% to 10%), suggesting it is not a complete solution for robust, real-world generalization.

**Questions:**

The proposed Policy Contrastive Decoding (PCD) method, which contrasts outputs from original ($o_i$) and object-masked ($\hat{o}_i$) observations, bears a strong conceptual resemblance to Classifier-Free Guidance (CFG) in diffusion models. In CFG, guidance is achieved by steering the generation process away from a null or unconditional prompt and toward a conditional prompt.

Given this parallel, have the authors considered implementing a more direct CFG-based approach for the diffusion-based policies (Octo and $\pi_0$)? Specifically, one could treat the object-masked input ${\hat{o_i}}$ as the "negative condition" (analogous to the unconditional prompt in standard CFG) and the original observation $o_i$ as the positive condition. The denoising network $\epsilon_{\theta}$ could then be guided using the standard CFG formulation, such as $\hat{\epsilon_{\theta}} = \epsilon_{\theta}(a_i^k, e_i, k) + w(\epsilon_{\theta}(a_i^k, e_i, k) - \epsilon_{\theta}(a_i^k, \hat{e}_i, k))$, where $\hat{e}_i$ is the embedding from the object-masked input and $w$ is a guidance scale.

This CFG-based approach would be a more native and direct way to apply guidance within the diffusion framework, potentially obviating the need for the proposed KDE-based Probabilistic Modeling (KDE-PM) scheme. Could the authors comment on whether they explored this alternative? If so, how did its performance and computational efficiency compare to the proposed PCD method? If not, do they foresee any challenges with such an implementation?

---

> ### Author Response · Authors · 2025-11-20
> **Response to Reviewer DdfD**
>
> We appreciate your insightful comments. We hope the following clarifications can address your concerns.
>
> **1) Different Optimal $\alpha$ Values for Different Baselines**\
> In our opinion, a single, uniform $\alpha$ is infeasible due to the vast architectural divergences across baseline polices, which span from autoregressive to diffusion models. It is noteworthy, however, that PCD (with any $\alpha > 0$) consistently outperforms the three baseline polices (see Figure 4(a)), which demonstrates that its efficacy is robust across a wide range of this hyper-parameter.
>
> **2) Lack of Training-Stage Solutions for Spurious Correlations**\
> Acknowledging this valuable feedback, our future work will focus on designing effective training-stage methods, intended to be complementary to PCD's advantages, to better solve the spurious correlation problem.
>
> **3) Lack of Zero-shot Results in Real-world Experiments**\
> It is well-established in the literature that the significant gap between pre-training datasets and real-world scenarios hinders pre-trained robot policies from achieving zero-shot generalization in the real world [1-3]. This is a fundamental limitation of existing robot policies themselves and falls outside the scope of what our method aims to correct.
> Furthermore, the results of our real-world experiments in Section 4.4 demonstrate the effectiveness of PCD in improving cross-domain zero-shot generalization.
>
> [1] O’Neill A, et al. Open X-Embodiment: Robotic learn. datasets and rt-x models. ICRA 2024.\
> [2] Li X, et al. Evaluating Real-World Robot Manipulation Policies in Simulation. CoRL 2024.\
> [3] Zhou Z, et al. Autoeval: Auto. Evaluation of generalist robot manipulation policies in the real world. arXiv:2503.24278, 2025.
>
> **4) Different Performance Gains in SIMPLER and Real-world Environments**\
> We attribute this primarily to two factors: **a) Evaluation Discrepancy**. The $\pi_0$ model in SIMPLER, ​separately trained on trajectories  of 73k from Fractal and 25k from Bridge, exhibits strong in-distribution generalization capacity, leaving minimal room for PCD to improve. Conversely, the $\pi_0$ model in real-world is trained on limited data, leading to finite performance and a higher susceptibility to spurious correlations. **b) Different $\pi_0$ Implementations**. The $\pi_0$ model used in SIMPLER is a third-party **PyTorch** implementation sourced from https://github.com/allenzren/open-pi-zero. In contrast, the $\pi_0$ model used for the real-world environment is the official **JAX** implementation from https://github.com/Physical-Intelligence/openpi.
>
> **5) The Absolute Success Rates in Hard Real-world Tasks Remain Low**\
> Our PCD functions as a refinement module to correct a policy's existing behaviors, rather than to generate novel skills for tasks on which the base model completely fails. That is to say, if the model lacks the fundamental capability to perform the task, merely focusing its attention on the correct target objects via PCD is insufficient to ensure completion.
>
> **6) Others**\
> Please refer to our **Response to Common Questions**.

---

> > ### Comment · Reviewer_DdfD · 2025-11-21
> > **Thank you for the response**
> >
> > Thank you for the response. I will keep my score.

---

> > > ### Author Response · Authors · 2025-11-25
> > > **Thanks**
> > >
> > > Thank you for your thoughtful feedback and active engagement throughout the rebuttal process. We sincerely appreciate your support and endorsement.

---

### Official Review · Reviewer_ReGE · 2025-11-07

**Soundness:** 3
**Presentation:** 2
**Contribution:** 2
**Rating:** 4
**Confidence:** 4

**Summary:**

The paper proposes Policy Contrastive Decoding (PCD), a training-free method that improves generalist robot policies at inference time by comparing the action distributions from the original observation with those from an object-masked observation. The object mask is obtained using a Track2Mask pipeline. The authors also introduce a KDE-based method to estimate action probabilities for diffusion policies. Experiments show that PCD improves OpenVLA, Octo, and Pi0 on SIMPLER simulation tasks, and Pi0 on a real-world pick-and-place task.

**Strengths:**

- The method is a training-free approach that can be applied to various generalist policies in a plug-and-play manner.
- The evaluation on both SIMPLER and real-world tasks shows improvement over the base policy.

**Weaknesses:**

- The method is mainly limited to simple pick-and-place tasks with a clear target object to mask, and it is hard to extend to more complex tasks that need long-horizon planning or involve multiple objects.

- The method introduces additional computation and latency, both from running Track2Mask and from the KDE-PM process. This is discussed in Appendix A.5. It may not be a major issue for the pick-and-place tasks studied in this paper, but it could be a limitation for tasks that require fast inference.

- In Table 1, the performance seems to vary depending on the object annotation strategy, and it is a sensitive design choice for each task and base policy. This requires the user to perform additional tuning to find the best setting.

**Questions:**

- At a high level, the method seems to be trying a similar idea as classifier-free guidance (CFG) in the diffusion model literature. Can the authors discuss more about this relation? In addition, can the authors try evaluating the diffusion policy with CFG instead of PCD + KDE-PM?  Since there is already the original observation $o$ and the object-masked observation $\hat{o}$, we can simply try a CFG-based action sampling by predicting the noise with a weighted combination of the two predictions
```
eps_uncond = model(x_t, t, \hat{o})
eps_cond   = model(x_t, t, o)
eps = eps_uncond + w * (eps_cond - eps_uncond)
```
- The paper mainly studies the use case for generalist policies, but will this method also work on smaller single-task policies trained on limited data? It might be interesting to train a smaller diffusion policy on the real-world dataset used in the experiments and test it out.
- Why does the performance differ so much between each object annotation strategy? What are the failure cases of Track2Mask?


I’d be happy to reconsider my score once these points are addressed.

---

> ### Author Response · Authors · 2025-11-20
> **Response to Reviewer ReGE**
>
> We appreciate your insightful comments. We hope the following clarifications can address your concerns.
>
> **1) Applicability to More Complex Tasks**\
> We respectfully clarify that PCD is not limited to single-object tasks and has been evaluated on complex scenarios.
>
> **a) Multi-Object Tasks**. Our method has already been tested on several multi-object tasks, including "Move Near", "Carrot Plate", and "Spoon Towel" in SIMPLER, and "Cookies Towel" and "Banana Plate" in the real world. Our strategy is to mask all task-relevant objects simultaneously. Inspired by your comments, we performed a new ablation study using the "Move Near" task. In the table below, for the task “Move A Near B”, we compare the results of i) masking only object A, ii) masking only object B, iii) masking A first, followed by masking B upon a successful grasp of A; and iv) masking both A and B. As can be observed, masking all task-specific objects simultaneously yields the best performance.
>
> | Method  | Task         | Mask A | Mask B | Mask A then B | Mask A and B Simultaneously |
> | ------- | ------------ | ------ | ------ | ------------- | ------------ |
> | OpenVLA | Move Near    | 60     | 50     | 57            | **62**           |
> | $\pi_0$ | Carrot Plate | 53     | 62     | 56            | **67**           |
>
> **b) Long-Horizon Tasks**.  Also, our PCD has been evaluated on the long-horizon task "Apple Drawer" in the SIMPLER environment. As illustrated in Appendix Table 3, the official language instruction of the task is composed of two sequential sub-tasks: "open the top drawer" and "place the apple into the top drawer". Our strategy is to mask objects based on the instructions of the current sub-task. For sub-task 1, we mask only the "top drawer"; for sub-task 2, we mask both the "apple" and the "top drawer". In other words, PCD holds potential for long-horizon tasks by leveraging an LLM-based planner with CoT reasoning [1, 2] to decompose abstract commands into concrete sub-tasks.
>
> [1] Huang et al. Thinkact: Vision-language-action reasoning via reinforced visual latent planning. NeurIPS 2025. \
> [2] Zawalski M, et al. Robotic Control via Embodied Chain-of-Thought Reasoning. CoRL 2024.
>
> We present a comprehensive analysis of this issue in **Appendix H** of the revised paper.
>
> **2) Performance Gaps among Different Object Annotation Strategies**\
> The observed performance gaps among different object annotation strategies arises from the distinct "preferences" inherent in each approach. For instance, prompts from humans versus those from object detection models exhibit different understandings of object presence. In the case of a "drawer", a human annotator tends to point to or box a specific drawer within a cabinet, whereas an object detection model typically bounds the entire cabinet containing all drawers. This leads to significant performance disparities on "drawer"-related tasks. We argue that no single object annotation strategy is universally optimal; rather, the most effective approach is contingent upon the specific task.
>
> Please refer to **Appendix I** of the revised paper for illustrative examples.
>
> **3) Effectiveness on Smaller Policies**\
> We thank the reviewer for this insightful suggestion. A key advantage of PCD is its training-free, plug-and-play capability. We concur that the method has the potential to enhance the performance of smaller models.
>
> To demonstrate this, we conducted a new real-world experiment on top of a Diffusion Policy. Specifically, we designed a "place bowl on plate" task using an AGILEX PIPER robotic arm, collected 50 demonstration trajectories and trained a Diffusion Policy from scratch. The model utilized a ResNet18 visual backbone and was implemented using the the LeRobot [1] framework. **Hyperparameters:** batch size=16, step=10000, optimizer=Adam, learning rate=1e-4
> scheduler=cosine, action chunk size=16, observation window=2.
>
> We seamlessly integrated PCD ($α$=1.0, $N$=24) into the baseline and conducted 20 evaluation trials. As shown in the table below, PCD significantly improves the performance of this smaller model.
> |Model |Success Rate (%)|
> | ----- | ------------ |
> |Diffusion Policy |15|
> |**+PCD**|**45**|
>
> Our code and demos for this experiment are publicly available at: https://github.com/PCD-robot/PCD-LeRobot.
>
> [1] https://github.com/huggingface/lerobot
>
> **4) Others**\
> Please refer to our **Response to Common Questions**.

---

> > ### Author Response · Authors · 2025-11-26
> > **Gentle Reminder**
> >
> > Dear Reviewer **ReGE**,
> >
> > We appreciate your insightful comments. We have revised our paper and provided a point-by-point response, and we hope it addresses your concerns.
> >
> > **We are very keen to clarify any remaining points and look forward to your feedback.**
> >
> > Best regards,
> >
> > Authors

---

### Author Response · Authors · 2025-11-20
**Response to Common Questions-Part II**

**3) Failure Cases of Track2Mask**\
From the Track2Mask pipeline presented in Appendix Fig.7, the failure cases fall into two categories: **a) Object Detection Failure**—the off-the-shelf open-vocabulary detector (i.e., GDINO) fails to localize objects in the initial observation; **b) Incomplete Object Masking**—the target objects along the trajectory are partially masked (please refer to **Appendix G** of the revised paper for illustrative examples) . According to Equation 2, when the first failure case occurs, PCD's prediction becomes equivalent to the original baseline result.

Motivated by your valuable suggestion, we conducted a new ablation study to assess the impact of the incomplete masking failure cases. The results are presented in the following table, where $\beta$ indicates the ratio of masked pixels manually excluded (the task is "pick coke can"). As can be seen, PCD's performance progressively declines as $\beta$ increases, until it approaches the baseline performance when $\beta$ reaches 60%. Nevertheless, both kinds of failure cases are exceptionally rare in our experiments, owing to the stability and effectiveness of the employed off-the-shelf models, i.e., GDINO for object detection and SAM v2 for object tracking and segmentation.

| Model| Baseline | +PCD ($\beta=0$) | $\beta=0.2$ | $\beta=0.4$ | $\beta=0.6$ |
| ------- | -------- | --- | ----------- | ----------- | ----------- |
| OpenVLA  | 25       | 40  | 32          | 28          | 27          |
| $\pi_0$  | 84       | 88  | 88          | 85          | 84          |

---

### Author Response · Authors · 2025-11-20
**Response to Common Questions-Part I**

We appreciate the reviewers for acknowledging the contributions of this work and all the constructive suggestions. We hope the following clarifications can address your common concerns.

**1) Computational Overhead**\
We concede that the performance improvements from our test-time PCD method incur an unavoidable additional computational overhead, a trade-off also observed in other training-free approaches [1,2,3].

The following table reports the computational overhead of three baseline policies integrated w/ or w/o PCD in the SIMPLER environment. **1) Inference Latency**. PCD approximately doubles the inference latency per step across all three policies. This is an expected trade-off, as PCD requires a second forward pass on the object-masked input to compute the contrastive signal. For Octo and $\pi_0$, although the KDE-PM process introduces additional inference time, the overall time overhead is still kept to roughly a twofold increase due to optimized parallel decoding and sharing of intermediate features and caches. Crucially, this increased inference time does not significantly affect the total task completion time in real-world scenarios, given the much longer duration of the robot's physical execution. As proven by Figure 3, PCD only brings **24\%** extra execution time on $\pi_0$. **2) Memory Cost**. The impact on memory cost varies based on the policy's architecture. For OpenVLA, which processes the original and masked inputs serially, the memory overhead is negligible, with the primary cost being the increased sequential processing time. In contrast, the memory costs of Octo and $\pi_0$ are $1.22 \times$ and $1.37 \times$ that of the baseline, respectively, due to their parallel data processing mechanisms. Please refer to **Appendix E** for a detailed analysis of PCD’s computational overhead.
| Model| OpenVLA | +PCD | Octo | +PCD | $\pi_0$ | +PCD |
| ------------------------------------------ | ------- | ------------ | ---- | --------- | ------- | ------------ |
| Average time cost for each infer. step (s) | 0.86    | 1.77| 0.21 | 0.39| 0.66| 1.09 |
| Memory cost (MB) | 16357   | 16869 | 2884 | 3528| 8535    | 11699 |

In future work, we plan to adopt more efficient parallel computing methods (e.g., Flash Attention [4]) and adaptively apply PCD to the most critical action steps to further reduce the additional overhead.

[1] Nakamoto M, et al. Steering Your Generalists: Improving Robotic Foundation Models via Value Guidance, CoRL 2025 \
[2] GR Team. Gemini Robotics: Bringing AI into the Physical World, arXiv 2025 \
[3] Wu M. ControlMLLM: Training-Free Visual Prompt Learning for Multimodal Large Language Models, NeurIPS 2024 \
[4] Dao et al. Flashattention: Fast and memory-efficient exact attention with io-awareness. NeurIPS 2022

**2) PCD vs. Classifier-Free Guidance (CFG)**\
The essential distinction lies in how and when the guidance is applied: CFG provides **implicit**guidance during the iterative decoding process. At each step of the diffusion, it steers the generation away from the unconditional prediction. This is an implicit probabilistic modeling process. In contrast, PCD provides **explicit** post-hoc corrections.

The quantitative comparison below shows that PCD significantly outperforms CFG across the nine SIMPLE tasks. However, we also observe that the time cost of PCD is $1.29 \times$ times that of CFG. Therefore, harnessing the complementary strengths of PCD and CFG to further address spurious correlations remains a promising avenue for future research.
| Model  | $\pi_0$  | +CFG     | +PCD     |
| :-------- | :------- | :------- | :------- |
| Average time cost for each inference step (seconds) | 0.66     | 0.84     | 1.09     |
| Average success rate over nine tasks (%)            | 63.9     | 62.7     | **68.1** |
| Close Drawer | **75.7** | **75.7** | 74.7     |
| Move Near    | **67.3** | 62.7     | 66.7     |
| Open Drawer        | 38.0     | 46.3     | **47.7** |
| Pick Coke     | 84.0     | 81.3     | **84.3** |
| Apple Drawer   | 17.0     | 13.0     | **26.0** |
| Carrot Plate    | 58.0     | 59.0     | **67.7** |
| Egg. Basket     | 86.0     | **87.3** | 83.7     |
| Spoon Towel   | 80.7     | 78.3     | **86.3** |
| Stack Cube   | 68.7     | 60.3     | **76.3** |

To further investigate CFG, we conduct a step-by-step analysis in the following table, applying CFG after a certain step and running 100 trials for each setting. The results show a clear trend: the earlier the CFG intervention, the more severe the performance degradation. The primary reason appears to be that CFG's influence on the noise space disrupts the semantic integrity of the model's reasoning process, ultimately degrading performance.
| Starting Step | Success Rate (%) |
| ---------- | -------- |
| 0          | 62.7             |
| 2          | 63.8             |
| 4          | 64.8             |
| 8          | 64.5             |

We have added this discussion to **Appendix F** of the revised paper.

---

### Author Response · Authors · 2025-11-30
**Summary by Authors**

Dear AC and Reviewers (ReGE, DdfD, gdDA),

We sincerely appreciate the valuable time and effort you have dedicated to reviewing our paper.

As highlighted by the reviewers, we propose a training-free and plug-and-play (**ReGE, DdfD, gdDA**) method to tackle an important  robot learning problem (**DdfD, gdDA**)—spurious correlations. The proposed method demonstrates flexibility and effectiveness
 (**ReGE, DdfD, gdDA**) through extensive experiments.

During the rebuttal phase, we addressed the concerns raised by all three reviewers regarding computational complexity, failure cases, method limitations and so on. We have also incorporated corresponding revisions into the updated paper.

**Before the OpenReview Data Leak incident** (officially confirmed on Nov 27, 2025), we were encouraged to receive positive feedback from reviewers DdfD and gdDA:
- **DdfD** confirmed that the concerns were resolved and maintained the initial score of **8** (Nov 21, 2025, 11:36).
- **gdDA**, upon further discussion, acknowledged that we addressed the vast majority of the concerns and raised the score from 4 to **6** (Nov 25, 2025, 00:09).

Regrettably, due to the suspension of author-reviewer communication following the data leak incident, we have not yet received a response from reviewer **ReGE**, despite his/her initial note stating: **"I’d be happy to reconsider my score once these points are addressed."**.
We would like to draw the AC's attention to our response regarding ReGE's concerns.
- **Shared Concerns**: ReGE's primary concerns regarding PCD's relation to CFG and failure cases were also raised by reviewers DdfD and gdDA. Given the positive endorsements from the two reviewers, we believe these issues have been resolved.
- **Specific Concerns**: Regarding ReGE's remaining questions, we provided detailed clarifications on the method's applicability to more complex tasks and the reasons behind performance gaps among different object annotation strategies. In addition, we demonstrated the method's effectiveness on smaller policies through new real-world experiments, and released the code and demos at https://github.com/PCD-robot/PCD-LeRobot.

We once again thank the AC and all reviewers for your hard work and valuable comments.

Best regards,

Authors

---

### Meta-Review · Area_Chair_RVfn · 2026-01-13

**Summary:**

This paper introduces Policy Contrastive Decoding (PCD), a training-free, plug-and-play inference-time method for mitigating spurious visual correlations in robotic foundation models. By contrasting action distributions from original observations against object-masked observations, enabled via an automatic Track2Mask pipeline and a KDE-based probabilistic formulation, PCD can be applied broadly to both autoregressive and diffusion-based policies without retraining. Reviewers consistently agree that the problem is important and that the method is flexible, model-agnostic, and empirically effective across simulation and real-world settings. Given the strength outweighs the drawbacks, ACs recommend acceptance.

**Reviewer Concerns:**

There are concerns about hyperparameter tuning, limits on tasks where the base policy lacks capability, and extra compute and latency overhead, failure cases, and the relation to CFG. Most of them were solved during rebuttal.

**Reviewer Scores:**

During rebuttal, the authors addressed most substantive concerns raised by reviewers. They provided a detailed computational overhead analysis, showing predictable but practically acceptable costs given real-world execution times. The authors also added new ablations and robustness studies, including analyses of masking failures, etc. Reviewer sentiment improved during discussion: one reviewer maintained a strong accept (8) after rebuttal, another raised the score from 4 to 6 after additional experiments and clarifications (revoked according to the authors).

---

### Decision · Program_Chairs · 2026-01-26

Accept (Poster)